# Lipocalin 13 enhances insulin secretion but is dispensable for systemic metabolic control

Lea Bühler[1,2,4], Adriano Maida[1,2,4], Elena Sophie Vogl[1,2,4], Anastasia Georgiadi[1,2,4], Andrea Takacs[1,2,4], Oliver Kluth[4,5], Annette Schürmann[4,5,6], Annette Feuchtinger[7], Christine von Toerne[8], Foivos-Filippos Tsokanos[1,2,4], Katarina Klepac[1,2,4], Gretchen Wolff[1,2,4], Minako Sakurai[1,2,4], Bilgen Ekim Üstünel[1,2,4], Peter Nawroth[2,4] (ORCID), Stephan Herzig[1,2,3,4] (ORCID)

Members of the lipocalin protein family serve as biomarkers for kidney disease and acute phase inflammatory reactions, and are under preclinical development for the diagnosis and therapy of allergies. However, none of the lipocalin family members has made the step into clinical development, mostly due to their complex biological activity and the lack of in-depth mechanistic knowledge. Here, we show that the hepatokine lipocalin 13 (LCN13) triggers glucose-dependent insulin secretion and cell proliferation of primary mouse islets. However, inhibition of endogenous LCN13 expression in lean mice did not alter glucose and lipid homeostasis. Enhanced hepatic secretion of LCN13 in either diet-induced or genetic obesity led to no discernible impact on systemic glucose and lipid metabolism, neither in preventive nor therapeutic setting. Of note, loss or forced LCN13 hepatic secretion did not trigger any compensatory regulation of related lipocalin family members. Together, these data are in stark contrast to the suggested gluco-regulatory and therapeutic role of LCN13 in obesity, and imply complex regulatory steps in LCN13 biology at the organismic level mitigating its principal insulinotropic effects.

## Introduction

Systemic energy homeostasis critically depends on the tightly controlled interaction between the central nervous system and peripheral organs. Classical endocrine circuits ensure the communication among metabolically active organs, thereby converting external environmental cues into metabolic responses at the cellular and molecular level. Consequently, endocrine mediators serve as prime targets for therapeutic intervention in various conditions of metabolic dysfunction, most notably exemplified by the use of the pancreatic hormone insulin for glucose control in diabetes or adrenal glucocorticoids for the treatment of inflammatory diseases.

Beyond the classical endocrine pathways, research over the past years has shown that almost every internal organ harbors an endocrine function, releasing numerous circulating factors with vastly uncharacterized tissue targets and/or receptors and thus an unexploited therapeutic potential.

As an important metabolic organ, the liver coordinates multiple aspects of systemic glucose, lipid, and protein metabolism, including the body's adaptation to the daily fasting–feeding cycle. In this respect, numerous liver-secreted factors, the so-called hepatokines, have been identified with both biomarker as well as regulatory functions in systemic energy homeostasis. This is exemplified by fetuin A, the circulating levels of which tightly correlate with obesity, diabetes, and other components of the metabolic syndrome in human cohorts (Ix et al, 2006; Stefan et al, 2008; Haukeland et al, 2012; Stefan & Haring, 2013).

Lipocalins represent a family of low molecular weight plasma proteins, secreted from various tissues and with diverse functions (Flower, 1996; Schlehuber & Skerra, 2005). Individual lipocalin family members have been discussed as therapeutic agents, for example, in immunomodulation, protection against oxidative stress, and amelioration of acute ischemic renal injury (Libert et al, 1994; Muchitsch et al, 1998; Lechner et al, 2001; Allhorn et al, 2002; Mishra et al, 2004; Mori et al, 2005). However, because of their rather complex biological activities and the lack of detailed cargo and receptor information in most instances, none of the lipocalins has reached clinical practice yet.

In this respect, lipocalin (LCN) 13 in particular has emerged as a potential (pre)clinical candidate for the treatment of metabolic dysfunction. Adenoviral delivery of a *Lcn13*-targeting siRNA into the liver abolished the positive effects on systemic glucose tolerance triggered by inhibition of TSC22D4 (Ustunel et al, 2016). Whole-body

[1]Institute for Diabetes and Cancer (IDC), Helmholtz Centre Munich, German Research Center for Environmental Health, Neuherberg, Germany    [2]Joint Heidelberg-IDC Transnational Diabetes Program, Inner Medicine I, Heidelberg University Hospital, Heidelberg, Germany    [3]Chair Molecular Metabolic Control, Medical Faculty, Technical University Munich, Munich, Germany    [4]German Center for Diabetes Research (DZD), Neuherberg, Germany    [5]Department of Experimental Diabetology, German Institute of Human Nutrition Potsdam-Rehbruecke (DIfE), Nuthetal, Germany    [6]Institute of Nutritional Science, University of Potsdam, Potsdam, Germany    [7]Research Unit Analytical Pathology, Helmholtz Centre Munich, German Research Center for Environmental Health, Neuherberg, Germany    [8]Research Unit Protein Science, Helmholtz Centre Munich, German Research Center for Environmental Health, Neuherberg, Germany

Correspondence: stephan.herzig@helmholtz-muenchen.de

transgenic overexpression of LCN13 in diabetic mice improved key parameters in glucose and lipid homeostasis (Cho et al, 2011; Sheng et al, 2011), and recombinant LCN13 enhanced insulin signaling in cultured adipocytes and muscle cells (Cho et al, 2011; Ustunel et al, 2016). Adenoviral transduction of *Lcn13* into obese animals suppressed hepatic gluconeogenesis and lipogenesis while stimulating fatty acid β-oxidation (Cho et al, 2011; Sheng et al, 2011), overall suggesting that hepatic LCN13 in particular may serve as a first candidate for further development of LCN13-centered compounds in the treatment of metabolic dysfunction. This notion was supported by the fact that LCN13 protein and RNA levels were diminished in both plasma and liver of diabetic animals as compared with healthy counterparts (Cho et al, 2011). Given the availability of lipid nanoparticle- or GalNac-based, liver-/hepatocyte-selective drug delivery systems, the hepatocyte-specific functions of LCN13 led to high expectations regarding its compatibility with today's highest safety and selectivity standards in current obesity and diabetes therapy. To circumvent the limitations and known inflammatory side effects of adenoviral gene transduction and zoom-in on the hepatocyte-selective LCN13 functions, we thus aimed at the vigorous preclinical evaluation of LCN13's therapeutic potential for metabolic disorders. Also, although the hepatic and systemic action of LCN13 have been studied to some degree already, a potential role of LCN13 for pancreatic islet function has not been investigated thus far, but given the critical importance of islet function for glucose control and the pathogenesis of diabetes, it is clearly needed to establish a comprehensive picture of LCN13's therapeutic capacities.

# Results

### LCN13 shows insulinotropic potential and drives islet cell proliferation ex vivo

To initially investigate the effects of LCN13 on β-cell function, we used the murine pancreatic β-cell line MIN6, which expresses a potential receptor for lipocalin family members (Blache et al, 1998). To guarantee optimal insulin secretion capacity, MIN6 cells were grown as pseudo-islets and used for experiments after reaching a size similar to primary mouse islets (Fig 1A; Hauge-Evans et al, 1999). To assess LCN13's ability to enhance glucose-stimulated insulin secretion (GSIS), pseudo-islets were incubated in low or high glucose (2.8 and 16.7 mM, respectively) in the presence or absence of recombinant bacterial LCN13 (Fig 1B). LCN13 increased GSIS in a strictly glucose-dependent manner. Importantly, LCN13 retained its insulinotropic effects also when using isolated primary mouse islets (Fig 1C). Insulin content of both MIN6 cells and primary islets remained constant in all treatment groups. Overall, recombinant bacterial LCN13 robustly induced GSIS in and ex vivo.

Given its positive effects on insulin secretion, we next investigated a potential role of LCN13 in islet cell proliferation. For this purpose, we studied whether LCN13 induced BrdU incorporation into isolated primary islet cell monolayers ex vivo. Indeed, compared with PBS-treated control cells, LCN13 induced islet cell proliferation more than twofold (Fig 1D). Taken together, LCN13 not only induced insulin secretion but also enhanced proliferation of primary islet cells ex vivo.

### Hepatic knockdown of LCN13 does not alter systemic glucose or lipid metabolism

To determine the source of systemic LCN13, we analyzed its expression levels in various tissue samples from lean C57BL/6N male mice. LCN13 expression was highly restricted to liver samples, verifying its role as a bona fide hepatokine (Fig 2A).

Next, we studied the physiological importance of hepatic LCN13 by its targeted knockdown (KD). This was achieved by hepatocyte-specific liponanoparticles (LNPs) carrying siRNA against *Lcn13* or luciferase as control. Using 0.5 mg/kg LNP was enough to achieve a marked LCN13 knockdown which was maintained over the course of 2 wk (Figs 2B and S1A). LNP treatment did not alter blood parameters of liver function, body, or liver weights (Fig S1B and C). Silencing hepatic *Lcn13* also had no effect on fasting glucose and insulin levels nor on the homeostatic model assessment of insulin resistance (HOMA-IR; Fig 2C). Glucose injection in fasted mice did not reveal any difference in glucose-dependent insulin secretion in vivo (Fig 2D). Accordingly, LCN13 KD mice did not exhibit changed glucose or insulin tolerance compared with mice receiving control LNPs (Fig 2E and F). Key parameters of lipid homeostasis in serum and liver also showed no alteration upon LCN13 KD (Fig 2G and H). In summary, targeted LCN13 KD did not lead to any changes in glucose or lipid handling in lean mice.

### Supra-physiological LCN13 plasma levels do not affect glucose regulation in lean mice

To further define the role of LCN13 under healthy conditions, we augmented systemic LCN13 levels in lean mice either by administration of an in-house produced recombinant mammalian LCN13 protein or by raising endogenous LCN13 levels using targeted hepatic overexpression (OE).

For the first experimental strategy, we produced LCN13 with a N-terminal Fc region (fragment crystallizable region of the human IgG heavy-chain) in CHO cells with a lone Fc tag as the control. Fc-LCN13 was successfully purified by immobilized metal ion affinity chromatography (IMAC) with a distinct peak in its elusion profile (Fig S2A). Enrichment and identity of Fc-LCN13 were confirmed by Coomassie-stained sodium dodecyl sulfate polyacrylamide gel electrophoresis (SDS–PAGE) gel (Fig S2B) and mass spectrometry (MS) analysis (Table S1). In addition, the Fc control and Fc-LCN13 proteins were verified by Western Blot, using commercially available α-IgG and α-LCN13 antibodies (Fig S2C). According to its amino acid sequence, the Fc-LCN13 has an expected size of 46 kD. However, in accordance with earlier findings, we detected multiple bands (Cho et al, 2011; Sheng et al, 2011). Bioactivity of Fc-LCN13 was confirmed by its potential to enhance glucose-dependent insulin secretion of MIN6 cells compared with the Fc control (Fig S2D). To test its metabolic regulatory potential in health, lean mice received PBS or different doses of the Fc control and Fc-LCN13 every second day for 12 d in total. Fc-LCN13 was detectable in the plasma of recipient mice (Fig 3A). Fasting glucose and insulin levels as well as HOMA-IR values were comparable in all mice (Fig 3B). Similarly, an IPGTT did not reveal any beneficial effects of Fc-LCN13 on glucose handling compared with the Fc-control (Fig 3C and D). Glucose

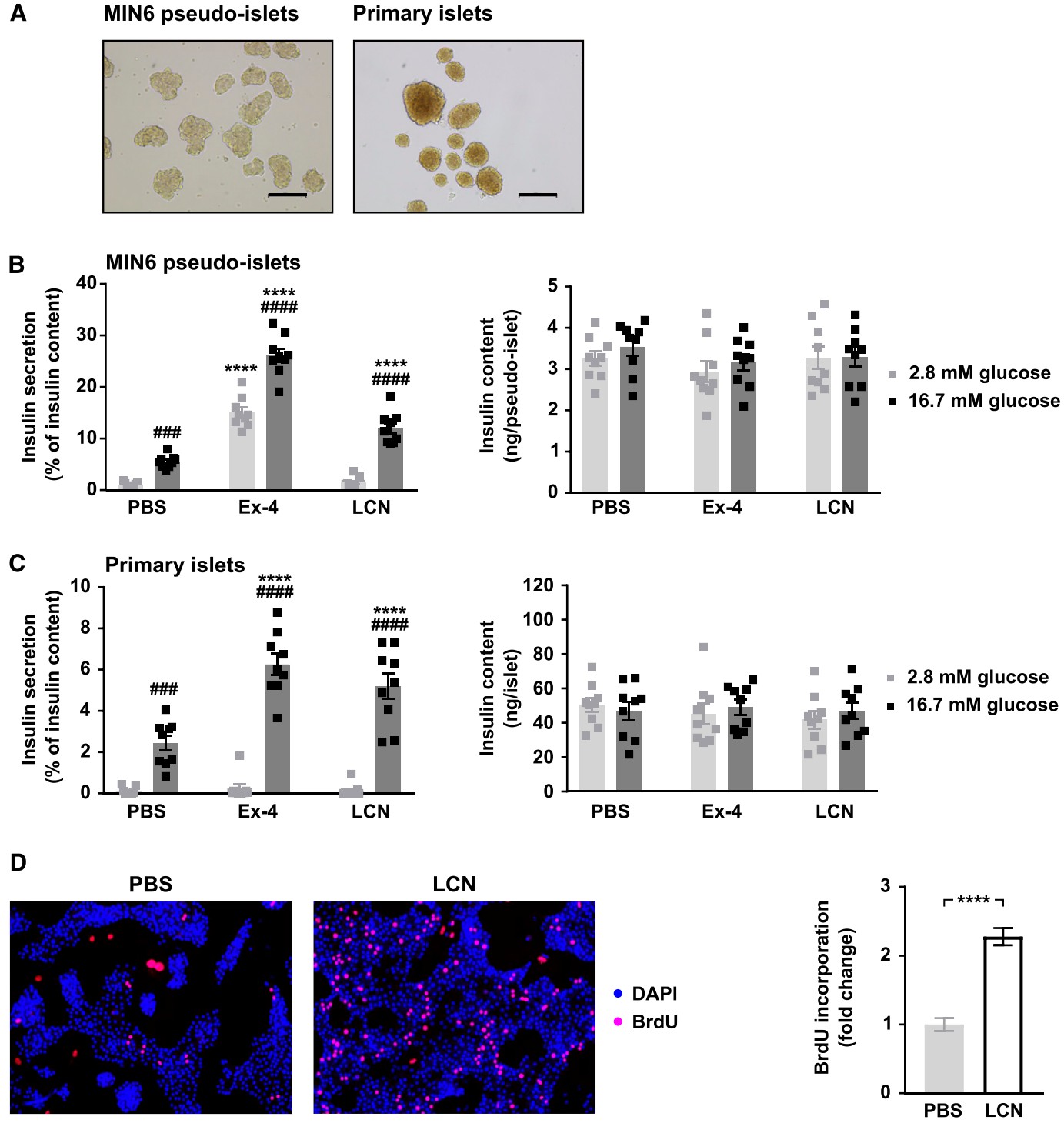

**Figure 1. LCN13 possesses insulinotropic effects and promotes islet cell proliferation ex vivo.**
**(A)** After 3 d in suspension culture MIN6 pseudo-islets reached a similar size as primary islets isolated from C57BL/6N male mice (8–12 wk). Scale bars: 500 $\mu$m. **(B, C)** Glucose-stimulated insulin secretion of (B) MIN6 pseudo-islets and (C) primary mouse islets (n = 3 independent experiments, each with three technical replicates per condition). Pseudo-islets and primary mouse islets were incubated with 10 nM recombinant bacterial LCN13 (LCN) for 12–24 h. PBS and 10 nM Exendin-4 (Ex-4) were used as negative and positive control, respectively. **(D)** Isolated primary mouse islets were grown as monolayer and incubated with PBS or 200 nM recombinant bacterial LCN13 (LCN) in the presence of 100 $\mu$M BrdU for 4 d. Representative pictures of stainings and the fold change of BrdU incorporation into islet cell nuclei (n = 3 independent experiments, each with 10–12 pictures per condition) are shown. Data information: All data are presented as mean ± SEM. [###]$P \leq 0.001$, [****]/[####]$P \leq 0.0001$. For glucose-stimulated insulin secretion data, two-way ANOVA was used. # represents significant difference between low and high glucose (Sidak's test), * represents significant difference between Ex-4/LCN and PBS (Dunnett's test). For proliferation data, unpaired, two-tailed t test was used.

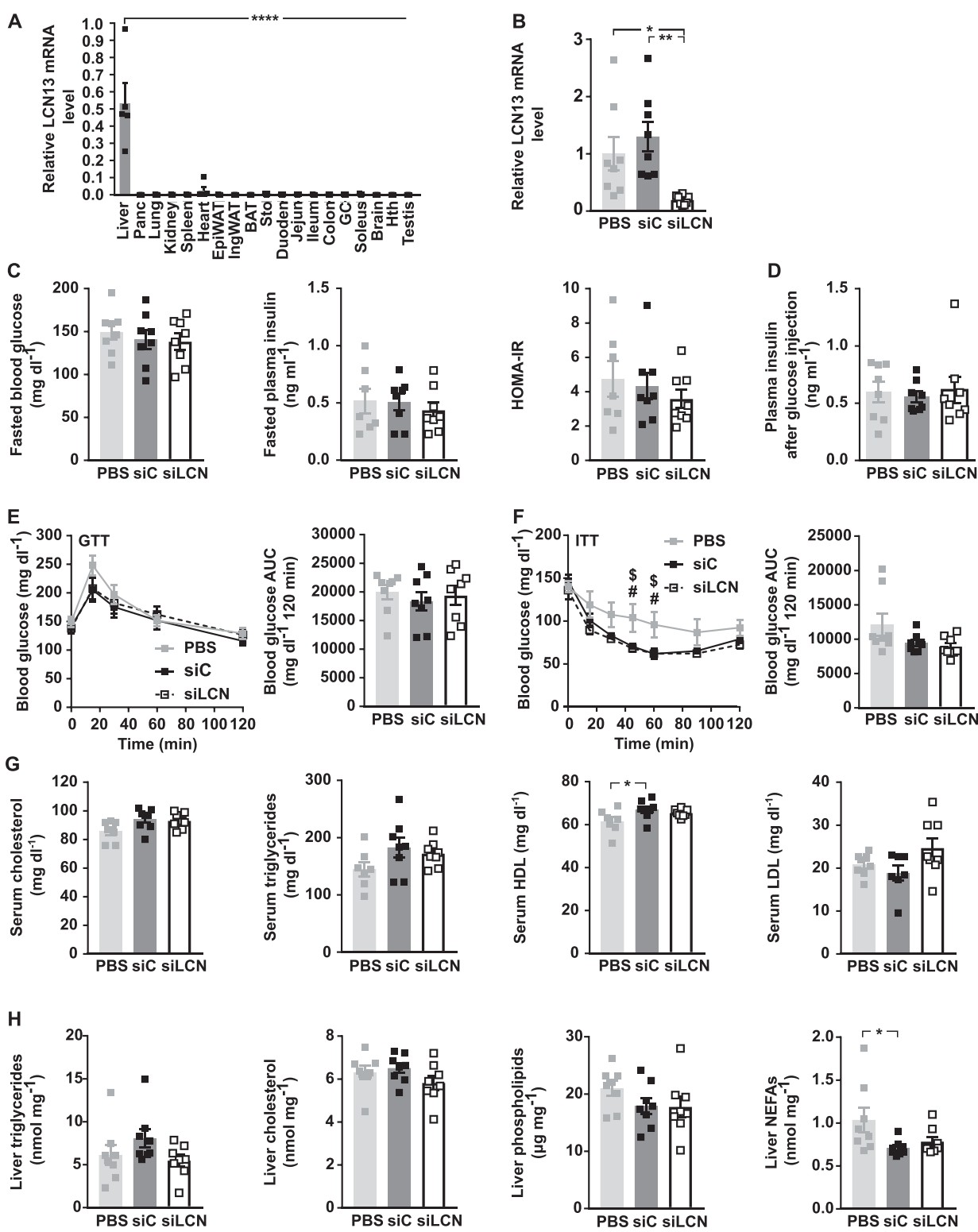

**Figure 2.  Liver-specific knockdown of LCN13 does not change glucose tolerance, key lipid parameters in serum or liver of lean mice.**
**(A)** Relative mRNA expression of LCN13 in liver, pancreas (Panc), lung, spleen, heart, epididymal white adipose tissue (EpiWAT), inguinal white adipose tissue (IngWAT), brown adipose tissue, stomach (Sto), duodenum (Duoden), jejunum (Jejun), ileum, colon, gastrocnemius muscle (GC), soleus, brain, hypothalamus (Hth), and testis of C57BL/6N male mice (13 wk; n = 5 mice for each group). **(B, C, D, E, F, G, H)** PBS or 0.5 mg/kg LNPs containing siRNA either against LCN13 (siLCN) or luciferase (siC) were i.v. injected in C57BL/6N male mice (8 wk; n = 8 mice for each group). **(B)** Relative mRNA expression of LCN13 in liver on day 15 after LNP administration. **(C)** Blood glucose, plasma insulin levels and HOMA-IR from 5-h fasted animals on day 7 after LNP administration. **(D)** Plasma insulin levels 15 min after 2 g/kg D-glucose were i.p. injected into

injection in fasted mice did not reveal any beneficial effects of Fc-LCN13 on glucose-dependent insulin secretion in vivo (Fig 3E).

For the second experimental strategy, we exploited adeno-associated virus (AAV)–mediated overexpression of LCN13 specifically in hepatocytes. The control AAV carried a plasmid with the sequence of a mutated, untranslated GFP. Successful LCN13 over-expression and secretion into the circulation was confirmed by Western blot analysis of plasma samples (Fig 3F). Instead of the predicted molecular weight of 18 kD, endogenous LCN13 was detected at multiple bands mimicking the observation of the Fc-LCN13. The applied technique was not sensitive enough to detect endogenous levels of LCN13 in control mice. Similar to mice receiving Fc-LCN13, LCN13 OE mice did not differ from control mice with regard to fasting glucose and insulin levels or HOMA-IR (Fig 3G). Likewise, insulin levels upon glucose injection were similar across all treatment groups (Fig 3H). All mice responded similarly to both glucose and insulin challenge (Fig 3I and J). However, as none of the mice markedly responded to the administered insulin dose, we cannot exclude that a potential insulin-sensitizing effect of LCN13 might have been missed (Fig 3J). Taken together, supra-physiological LCN13 levels did not influence glucose handling in healthy, lean mice.

### Hepatocyte-borne LCN13 does not affect systemic glucose and lipid metabolism in mice with diet-induced obesity

As we did not see any discernible difference between LCN13 OE and control mice in a healthy condition, we next studied the importance of LCN13 in mice with impaired metabolism. First, we assessed the therapeutic potential of LCN13 in a mouse model of diet-induced obesity. For this, AAV-mediated LCN13 overexpression was induced, after mice had been fed with a high fat diet (HFD) (60% of calories from fat) over the course of 14 wk. Pronounced LCN13 over-expression was confirmed both in liver and in plasma of LCN13 OE mice (Fig 4A and B). Of note, HFD alone did not alter LCN13 expression compared with the control low fat diet (10% of calories from fat; Fig 4A). Parameters of liver function, body, and liver weights were comparable between LCN13 OE mice and control mice injected either with PBS or the control AAV (Fig S3A and B). Levels of blood glucose and plasma insulin in the fasted state were not significantly altered upon LCN13 OE (Fig 4C and D). LCN13 OE mice were significantly less insulin resistant (lower HOMA-IR) than PBS-injected mice after 3 wk, but not after 14 wk (Fig 4E). LCN13 OE mice did not show improved glucose nor insulin tolerance compared with control mice at different time points after AAV administration (Fig 4F and G). However, as none of the mice markedly responded to the administered insulin dose, we cannot exclude that a potential insulin-sensitizing effect of LCN13 might have been missed (Fig 4G). Supporting a negligible effect of LCN13 on systemic metabolism in obese mice, acute and long-term blood glucose values of ad-

libitum fed mice were comparable across all treatment groups (Fig S3C). LCN13 OE also had no noticeable effect on serum lipid parameters (Fig 4H). Given its positive effects on islet cell proliferation in vitro, we assessed whether hepatic LCN13 overexpression promoted islet cell proliferation in this in vivo study. For this, mice received i.p. BrdU injections on three consecutive days, before pancreata were collected and immunofluorescence stainings were performed (Fig S3D). Higher LCN13 levels did not affect the total amount of insulin and glucagon positive cells in the analyzed pancreas sections (Fig S3E). Similarly, the amount of proliferating β- (insulin positive) and α- (glucagon positive) cells did not change upon LCN13 overexpression (Fig S3F). In line with this, the average insulin and glucagon amount per islet, mean islet size and total islet area remained unchanged in mice receiving the LCN13 over-expressing AAV compared with the AAV control group (Fig S3G–I). In summary, elevated LCN13 levels did not influence islet cell composition or proliferation in HFD-fed mice.

Next, we assessed the potential of LCN13 to blunt metabolic dysfunction in mice fed a HFD. For this, LCN13 was overexpressed in lean mice on chow diet and HFD feeding was initiated 5 wk later (Fig 5A). LCN13 overexpression was validated both in liver and plasma of LCN13 OE mice (Figs 5B and 3F). Importantly, HFD feeding alone did not alter LCN13 expression compared with chow-fed mice (Fig 5B). AAV-administration had no detrimental effect on liver integrity (Fig S4A). Body and liver weight were comparable across the treatment groups (Figs 5A and S4B). Blood glucose and plasma insulin levels in the fasted state as well as the calculated insulin resistance (HOMA-IR) were similar in all mice (Fig 5C–E). LCN13 OE mice responded equally to both glucose and insulin challenge than control mice (Fig 5F and G). Furthermore, ad-libitum fed blood glucose as well as HbA1c values did not differ upon LCN13 OE (Fig S4C). Higher LCN13 levels in the plasma did also not show any consistent effect on serum lipid parameters compared with both PBS and the AAV control group (Fig 5H). In summary, targeted hepatic LCN13 over-expression did not improve systemic glucose and lipid metabolism, neither in a therapeutic nor in a preventive setting.

### Hepatic LCN13 overexpression does not ameliorate genetically induced metabolic impairments

To verify our findings on the negligible role of LCN13 in systemic metabolism, we additionally used the leptin receptor deficient db/db mouse, a genetic mouse model of obesity (Chen et al, 1996). db/db mice showed a consistent, although not significant trend towards reduced LCN13 mRNA expression in the liver compared with their wild-type counterparts (Fig 6A). LCN13 overexpression in db/db mice led to significantly increased mRNA and protein levels in the liver and circulation, respectively (Fig 6A and B). Similar to the previous experiments, LCN13 overexpression did neither change

5 h fasted animals on day 7 after LNP administration. **(E)** IPGTT (2 g/kg D-glucose) after 5 h of fasting on day 7 after LNP administration. **(F)** IPITT (1 U/kg insulin) after 5 h of fasting on day 13 after LNP administration. **(G)** Cholesterol, triglyceride, high-density lipoprotein, and low-density lipoprotein levels in serum on day 15 after LNP administration. **(H)** Triglyceride, cholesterol, phospholipid, and non-esterified fatty acid levels in liver on day 15 after LNP administration. Data information: All data are presented as mean ≤ SEM. *$^{/\#/\$}P$ ≤ 0.05; **$P$ ≤ 0.01; ****$P$ ≤ 0.0001. For both IPGTT and IPITT two-way ANOVA (Tukey's test) and for all other data sets one-way ANOVA (Tukey's test) was used. In (F), # represents significant difference between PBS and siLCN treatment, whereas $ represents significant difference between PBS and siC treatment.

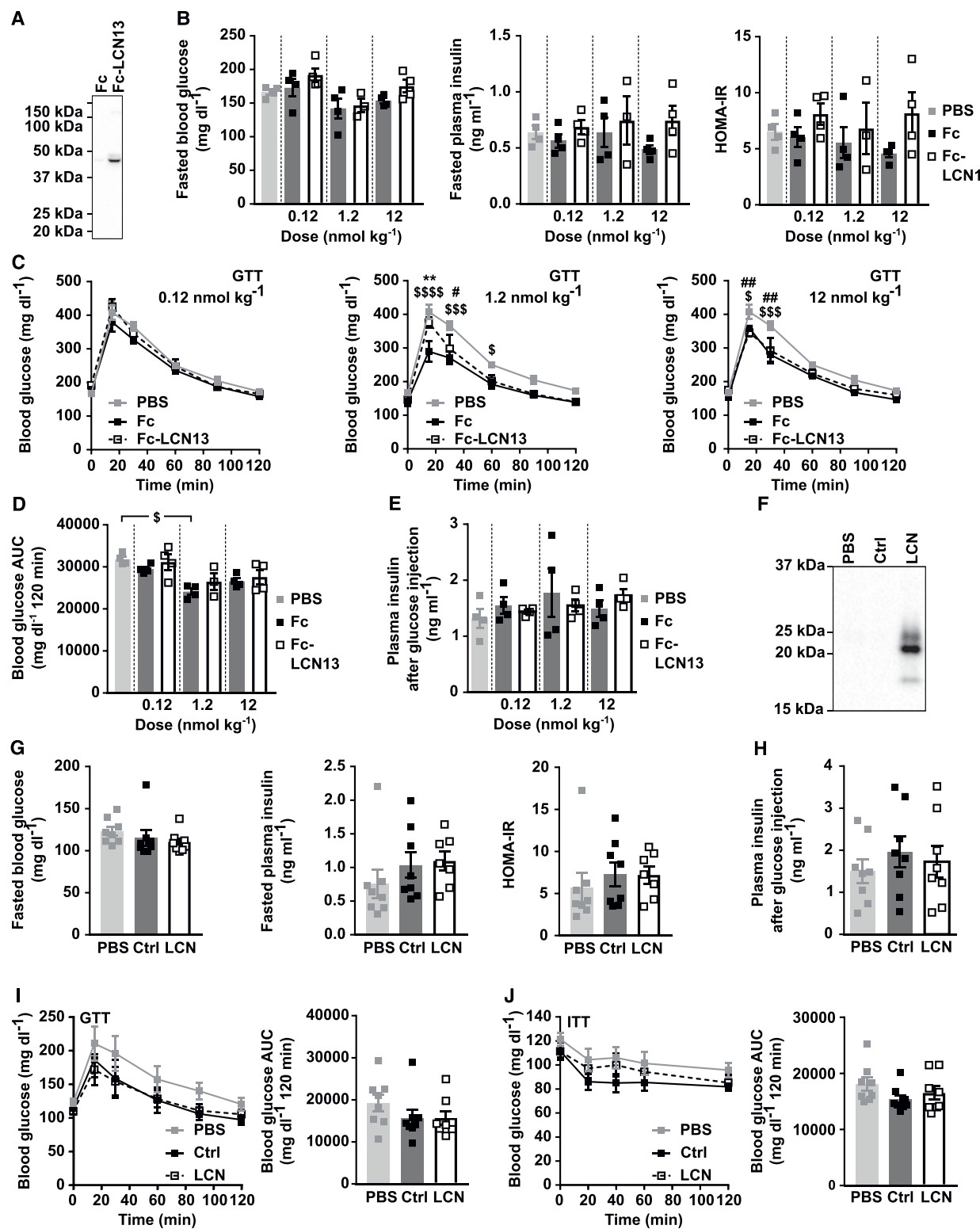

**Figure 3. Increased LCN13 plasma levels do not affect glucose handling in lean mice.**
**(A, B, C, D, E)** C57BL/6N male mice (11 wk) received different doses of Fc control or Fc-LCN13 via intraperitoneal injections every second day for 12 d in total (n = 4 mice for each group). **(A)** Representative immunoblot of Fc-LCN13 in plasma of mice injected with 1.2 nmol/kg of the recombinant proteins on day 12 using αLCN13. **(B)** Blood glucose, plasma insulin levels, and HOMA-IR from 5 h fasted animals on day 12. **(C, D)** IPGTT (2 g/kg D-glucose) and (D) the corresponding glucose area under the curve (AUC) after 5 h of fasting on day 12. **(E)** Plasma insulin levels 15 min after 2 g/kg D-glucose were i.p. injected into 5-h fasted animals on day 12. **(F, G, H, I, J)** PBS or adeno-associated viruses (AAVs) overexpressing either LCN13 (LCN) or a mutated, untranslated GFP (Ctrl) were administered to C57BL/6N male mice (7 wk) at a dose of 5 × 10$^{11}$ vg/

serum parameters of liver function nor ad libitum blood glucose, HbA1c levels, body or liver weights (Fig 6C–E). All mice had similar fasted blood glucose, fasted plasma insulin, and calculated HOMA-IR values (Fig 6F). Furthermore, all mice responded similarly to glucose and insulin challenges (Fig 6G and H). However, as none of the mice markedly responded to the administered insulin dose, we cannot exclude that a potential insulin-sensitizing effect of LCN13 might have been missed (Fig 6H). LCN13 OE mice did not exhibit any difference in serum lipids compared with control db/db mice (Fig 6I).

Examining pancreas sections revealed that some db/db mice exhibited significantly different islet cell compositions (Fig S5A and B; depicted as triangles). Islets from these mice had 35% insulin positive and 51% glucagon positive cells which drastically differed from the rest of the db/db mice (77% insulin positive and 6% glucagon positive cells). This different islet cell composition was accompanied by significantly higher glucagon expression per islet and significantly lower mean islet size and overall islet area (Fig S5D–F). With one exception, this further progressed decrease in β cell area correlated with higher blood glucose AUC in the conducted glucose tolerance test (Fig 6G). Of note, regardless whether the corresponding data of this particular set of mice were included into the analysis or not, LCN13 overexpression did not show any beneficial effect on islet cell proliferation (Fig S5C) or any other of the analyzed parameters in db/db mice. In summary, reminiscent of HFD-feeding, genetically induced metabolic dysfunction of db/db mice was not ameliorated by targeted hepatic LCN13 overexpression.

One common feature of all preceding experiments was the pronounced higher level of circulating LCN13, irrespective of whether we used recombinant Fc-LCN13 or the LCN13 overexpressing AAV compared with control mice (Figs 3A and F, 4A and B, 5B, and 6A and B). To ensure that these supra-physiological amounts did not impair LCN13's function, we performed a titration experiment of the LCN13 overexpressing AAV in vivo. $5 \times 10^9$, $5 \times 10^{10}$ and $5 \times 10^{11}$ vg/mouse resulted in hepatic LCN13 mRNA levels spanning two orders of magnitude (~5–500 times higher than control wild type mice; Fig 7A), which was also apparent at the plasma LCN13 level (Fig 7B). None of the viral doses affected liver integrity, body or liver weight (Fig 7C and D). There was no difference in fasting blood glucose, plasma insulin or the HOMA-IR (Fig 7E). In line with this, glucose and insulin tolerance as well as serum lipid parameters were comparable across all treatment groups (Fig 7F–H). Therefore, we concluded that the pronounced increase in systemic LCN13 in all preceding experiments was not the reason why LCN13 did not appear as beneficial metabolic regulator.

### Expression of related lipocalins is not coupled to hepatic LCN13 levels

Besides LCN13, a number of other lipocalin family members, such as LCN2, LCN5, LCN14, apolipoprotein D (ApoD), apolipoprotein M (ApoM), major urinary protein 1 (MUP1), prostaglandin D2 synthase (PTGDS), and retinol-binding protein 4 (RBP4), have been indicated in contributing towards systemic energy homeostasis. Thus, we investigated whether the expression of these lipocalins was affected by the manipulation of hepatic LCN13 levels which could have obscured LCN13's metabolic function. As the amino acid sequence of lipocalins are known to be rather dissimilar (<20%; Flower, 1996), we also included LCN14, LCN3, LCN4, and PTGDS based on their unusual high sequence similarity with LCN13 (65%, 56%, 48%, and 39%, respectively). Targeted knockdown of hepatic LCN13 in lean mice was accompanied by significantly higher LCN2 levels in epididymal white adipose tissue (Fig 8A). Apart from this, none of the investigated lipocalins showed differential expression in the analyzed tissues. Targeted overexpression of hepatic LCN13 in db/db mice did not affect the expression of any of the analyzed lipocalins in liver, white adipose tissue or muscle (Fig 8B). Interestingly, although AAV-mediated LCN13 overexpression was under the control of the hepatocyte-specific promoter LP1, we found a small, significant up-regulation of LCN13 also in the inguinal white adipose tissue (Fig 8B). Taken together, manipulation of hepatic LCN13 levels, either by targeted knockdown or overexpression, did not lead to a consistent change in expression levels of related lipocalin family members.

Overall, our preclinical experiments demonstrated that despite the potential capacity of LCN13 to boost pancreatic β-cell insulin secretion and proliferation, its hepatic function is not sufficient to trigger any significant systemic metabolic phenotype, including fasting glucose and insulin levels, glucose and insulin tolerance, and lipid homeostasis. Our data rather discourage the further clinical development of LCN13 peptides and/or agonists for metabolic dysfunction as associated with obesity and diabetes.

## Discussion

The inability of existing anti-diabetic drugs to maintain glycemic control, as well as the substantial side effects, reinforce the dire need for new therapeutic solutions for T2D patients. Inter-organ hormonal circuits might provide effective leveraging points for such future T2D treatments. In this context, liver-secreted LCN13 has emerged as a potent insulin-sensitizer whose expression is reduced in obese mice and T2D patients (Cho et al, 2011; Sheng et al, 2011; Ustunel et al, 2016). Here, we thoroughly evaluated LCN13's potential to serve as a biomarker and therapeutic target for metabolic disorders.

Although LCN13 consistently promoted glucose-dependent insulin secretion and islet cell proliferation in vitro, manipulation of its hepatic expression levels in lean or obese mice did neither affect systemic glucose and lipid metabolism, nor β cell proliferation in vivo. These results contradict previously published data which

mouse via the tail vein (n = 8 mice for each group). **(F)** Representative immunoblot of LCN13 in plasma 4 wk after AAV administration. **(G)** Blood glucose, plasma insulin levels and HOMA-IR from 5 h fasted animals 4 wk after AAV administration. **(H)** Plasma insulin levels 15 min after 2 g/kg D-glucose were i.p. injected into 5 h fasted animals 4 wk after AAV administration. **(I)** IPGTT (2 g/kg D-glucose) after 5 h of fasting and 4 wk after AAV administration. **(J)** IPITT (0.75 U/kg) after 5 h of fasting and 5 wk after AAV administration. Data information: All data are presented as mean ± SEM. $^{#/\$}P \leq 0.05$; $^{**/##}P \leq 0.01$; $^{\$\$\$}P \leq 0.001$; $^{\$\$\$\$}P \leq 0.0001$. For both IPGTT and IPITT two-way ANOVA (Tukey's test) and for all other data sets, one-way ANOVA (Tukey's test) was used. In (C, D), * represents significant difference between Fc and Fc-LCN13, # represents significant difference between PBS and Fc-LCN13, whereas $ represents significant difference between PBS and Fc.
Source data are available for this figure.

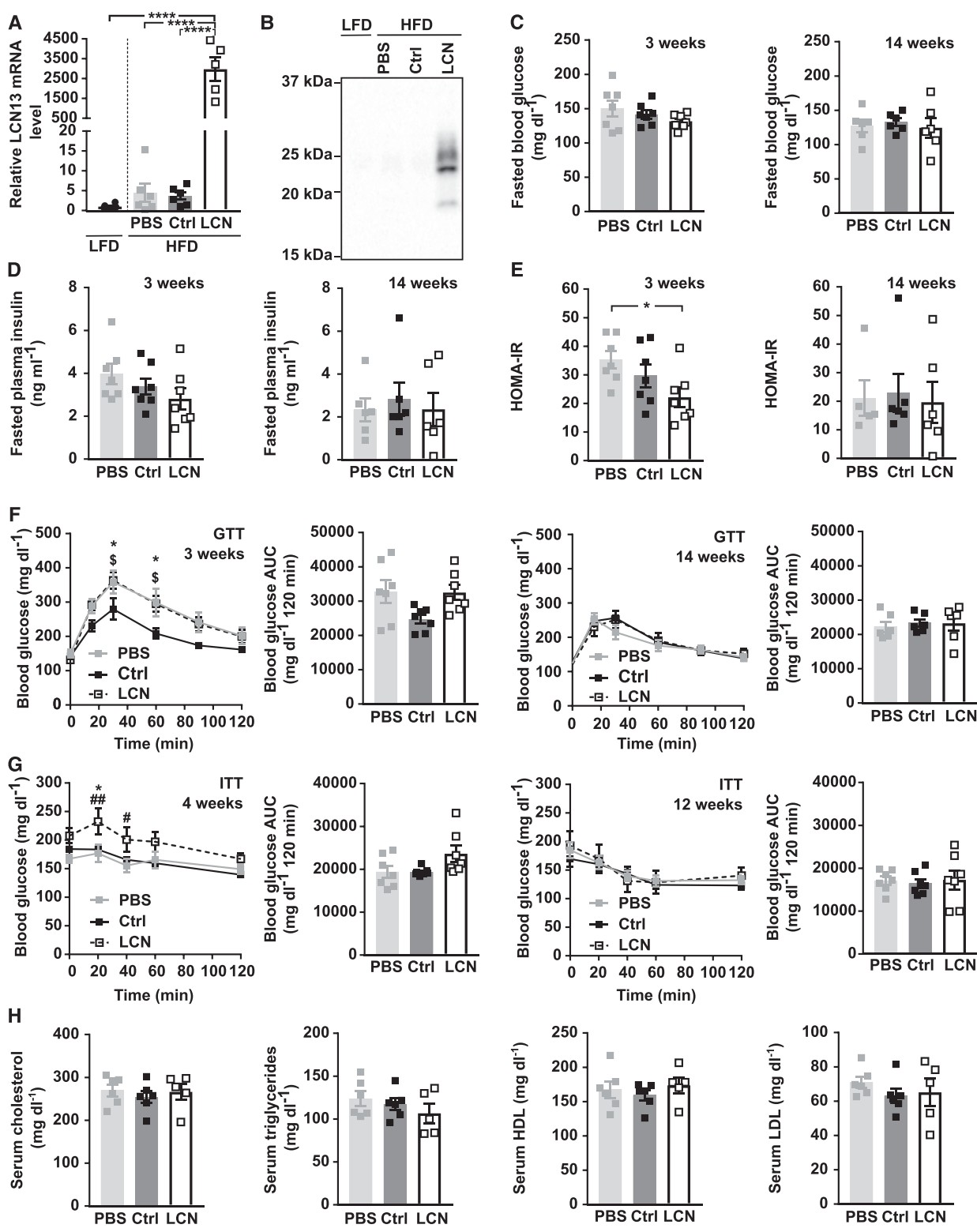

**Figure 4. Hepatic LCN13 overexpression does not alter metabolic dysfunction in high fat diet-fed mice.**
Diet-induced obesity was evoked by feeding C57BL/6N male mice (5 wk) a high-fat diet (60% calories from fat). Control mice received a low-fat diet (10% calories from fat). PBS or adeno-associated viruses (AAVs) overexpressing either LCN13 (LCN) or a mutated, untranslated GFP (Ctrl) were administered after 14 wk at a dose of $5 \times 10^{11}$ vg/mouse via the tail vein (n = 5–7 mice for each group). **(A)** Relative mRNA expression of LCN13 in liver 16 wk after AAV administration. **(B)** Representative immunoblot of LCN13 in plasma 3 wk after AAV administration. **(C, D, E)** Blood glucose, (D) plasma insulin levels, and (E) HOMA-IR from 16 h fasted animals 3 and 14 wk after AAV administration. **(F)** IPGTT (1 g/kg D-glucose) after 16 h of fasting 3 and 14 wk after AAV administration. **(G)** IPITT (1 and 1.5 U/kg insulin, respectively) after 5 h of fasting 4 and 12 wk after AAV

identified LCN13 as systemic metabolic regulator (Cho et al, 2011; Sheng et al, 2011; Ustunel et al, 2016). While Cho et al (2011) and Sheng et al (2011) observed reduced LCN13 levels both in liver and the circulation in different mouse models of obesity (Cho et al, 2011; Sheng et al, 2011), LCN13 expression was unresponsive to metabolic dysfunction in our hands (Figs 4A, 5B, 6A, and 7A).

State-of-the-art gene delivery systems allowed us to minimize inflammatory side effects, which distinguishes this work from previous studies. It is conceivable that LCN13 reveals its metabolic function exclusively in the presence of certain inflammatory signals, such as TNF-α, INF-γ, IL-1β, or IL-6 which could be essential for the expression of its cargo, receptor or downstream signaling molecules. Here, targeted silencing of LCN13 was, for instance, achieved by the administration of hepatocyte-specific LNPs. Transferred siRNA molecules were modified (2′-O-methylation and phosphorothioate linkages) to generate non-inflammatory oligonucleotides with optimal stability (Judge et al, 2006; Selvam et al, 2017). In line with this, we used a pseudo-typed AAV2/8 virus together with a LP1 promoter to achieve low immunogenicity while increasing LCN13 expression precision in hepatocytes (Cheng et al, 1993; Gao et al, 2002; Nakai et al, 2005; Wang et al, 2005; Akache et al, 2006; Nathwani et al, 2006). In general, AAVs are known to be non-pathogenic and to evoke highly transient and very mild immune responses only (Chirmule et al, 1999; Hernandez et al, 1999; Zaiss & Muruve, 2008). Studies conducted by Cho et al (2011), Sheng et al (2011), and Üstünel et al (2016), in contrast, exploited first generation adenoviruses, which induce prominent innate and adaptive immune responses, as means to overexpress or silence LCN13 (Zaiss et al, 2002; Liu & Muruve, 2003; McCaffrey et al, 2008). In addition, we manufactured LCN13 in mammalian cells, instead of bacteria, to prevent the contamination with bacterial products such as LPS.

Besides using adenovirus-mediated overexpression, Cho et al (2011) and Sheng et al (2011) also generated a LCN13 transgenic mouse line characterized by improved glucose handling, insulin responsiveness and ameliorated hepatic steatosis upon HFD treatment (Cho et al, 2011; Sheng et al, 2011). The chosen CAG (CMV enhancer, chicken β-Actin promoter and rabbit β-Globin splice acceptor site) promoter is already active in the unfertilized ovum and later in almost every cell type (Niwa et al, 1991; Okabe et al, 1997). This makes differentiating developmental from purely metabolic effects and the investigation of LCN13's hepatokine function challenging. However, the authors additionally reported that injection of anti-LCN13 serum worsened glucose homeostasis and induced lipogenic gene transcription while inhibiting fatty acid β-oxidation in lean mice (Sheng et al, 2011). Thus, the distinct methods used for manipulating LCN13's expression in this and previous studies cannot be the only reason for the contrasting results.

In general, mouse strain and age could both be further underlying factors for poor reproducibility. Whereas in our previous work db/db mice on a BKS background (BKS.Cg-Dock7m +/+ Lepr^db/J [000642]) were used (Ustunel et al, 2016), all mice in this and the studies of Cho et al (2011) and Sheng et al (2011) were on a C57BL/6 background. Therefore, genetic differences are unlikely the reason for the observed differences. Concerning age, some of our data are derived from mice which were markedly older and, therefore, possibly displayed an enhanced disease progression compared with the previous studies. In the study of Cho et al (2011), db/db mice which received the control adenovirus, showed a HOMA-IR value of ~25 at the age of 10 wk (fasted for 16 h; blood glucose ~120 mg/dl, plasma insulin ~3.4 ng/ml; Cho et al, 2011). In comparison, db/db mice which were injected with the LCN13 overexpressing AAV in our study had an average HOMA-IR value of 50 (fasted for 16 h; blood glucose ~244 mg/dl, plasma insulin ~4.1 ng/ml; Fig 6A) at the age of 17 wk. This higher degree of metabolic dysfunction could have possibly been accompanied by compromised tissue responsiveness towards LCN13 in our study. However, the metabolic indifference upon LCN13 KD in healthy lean mice (Fig 2) shows that distinct disease stages also cannot be the sole explanation for the observed lack of reproducibility.

Another source of variation could be the diet, as both chow diet and HFD used in this and previous studies varied with respect to composition. Fat and fiber content (% of mass) in this and our previous study were more akin than the one used in the publications by Cho et al (2011) and Sheng et al (2011) (This study: 5.1% fat, 4.5% fiber; Üstünel et al [2016]: 4.3% fat, 4.7% fiber; Cho et al [2011] and Sheng et al [2011]: 9% fat, 2.4% fiber). In addition, all chow diets were purchased from different vendors. Whereas Cho et al (2011) and Sheng et al (2011) used a 45% HFD, we fed our mice with a 60% HFD from the same supplier. Therefore, mice on HFD used in this study, had a potentially further progressed disease stage with more severe metabolic endotoxemia. Albeit the source of lipids in both HFDs was the same, the composition varied slightly (45% HFD: 88% lard, 12% soybean oil; 60% HFD: 91% lard, 9% soybean oil). Taken together, these food variables could have not only influenced the degree of metabolic endotoxemia and thus insulin resistance (Cani et al, 2007), but also the identity and availability of LCN13's putative, lipophilic cargo.

Differences in mouse husbandry could also be an underlying factor for the observed discrepancies. Most importantly divergent hygienic measurements and barrier facilities together with the already discussed differences in age, disease stage and diet could have led to distinct gut microbiota (De Filippo et al, 2010; Hufeldt et al, 2010; Franklin & Ericsson, 2017). Microbial imbalance and endotoxemia are able to promote the expression of pro-inflammatory cytokines, such as IL6, TNF-α, IL-1, and plasminogen activator inhibitor-1 in a TLR-4 dependent manner (Cani et al, 2007). Thus, it could be possible that distinct gut microbiota led to varying levels of inflammation and insulin resistance which might have differently affected LCN13 responsiveness in this and previous studies.

Lipocalins, such as LCN5, LCN13, LCN14, and Mup1, can have similar metabolic functions (Hui et al, 2009; Zhou et al, 2009; Cho et al, 2011; Sheng et al, 2011; Ustunel et al, 2016; Lee et al, 2016; Seldin

administration. **(H)** Cholesterol, triglyceride, high-density lipoprotein and low-density lipoprotein levels in serum 16 wk after AAV administration. Data information: All data are presented as mean ± SEM. */#/$P ≤ 0.05; ##P ≤ 0.01; ****P ≤ 0.0001. For both IPGTT and IPITT two-way ANOVA (Tukey's test) and for all other data sets one-way ANOVA (Tukey's test) was used. In (F, G), * represents significant difference between Ctrl and LCN13. # represents significant difference between PBS and LCN. $ represents significant difference between PBS and Ctrl.
Source data are available for this figure.

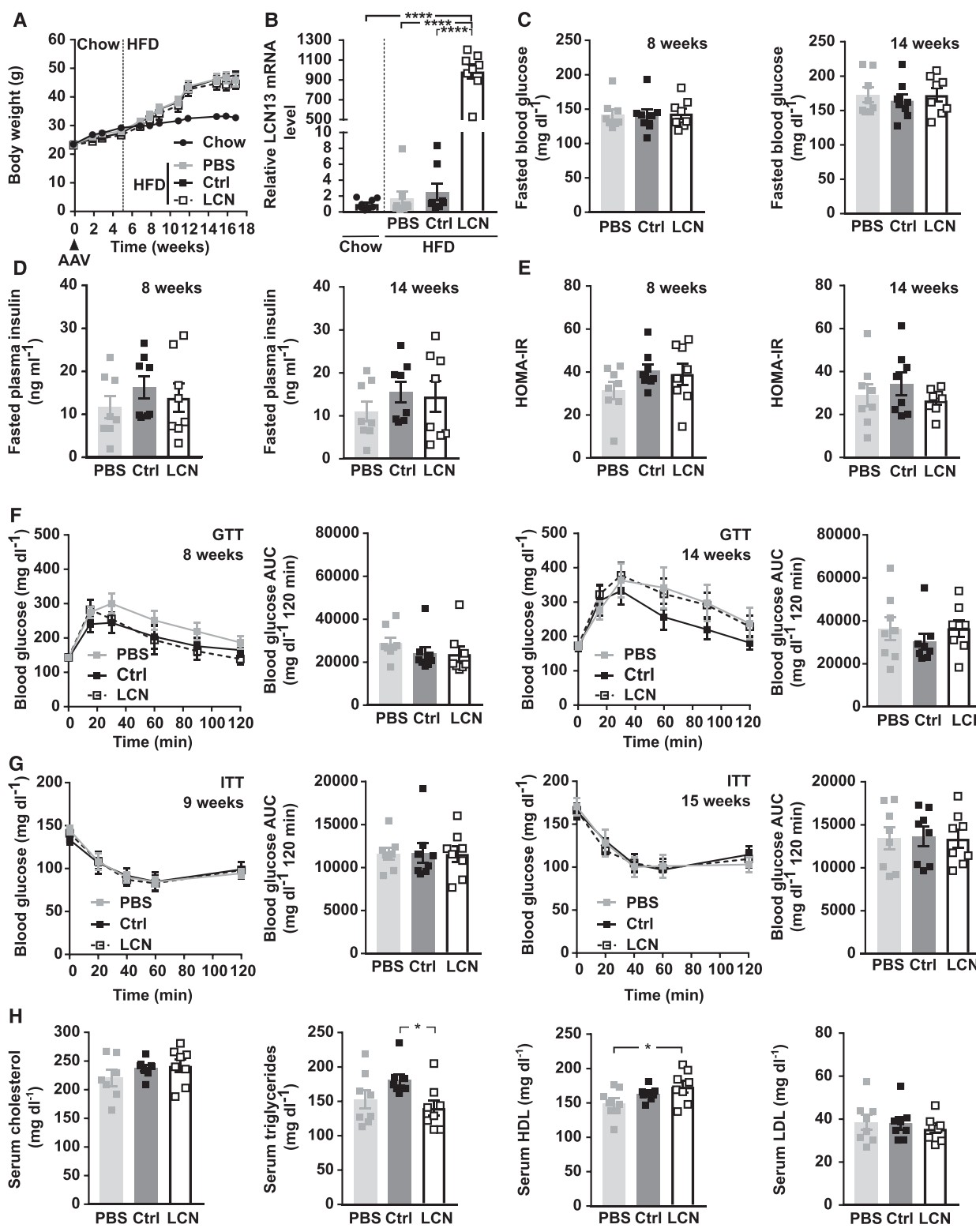

**Figure 5. Hepatic LCN13 overexpression does not prevent diet-induced obesity or related metabolic impairments.**
PBS or adeno-associated viruses (AAVs) overexpressing either LCN13 (LCN) or a mutated, untranslated GFP (Ctrl) were administered to C57BL/6N male mice (7 wk) at a dose of 5 × 10¹¹ vg/mouse via the tail vein (n = 8 mice for each group). The chow diet was replaced by a high fat diet (60% of calories from fat) 5 wk later. Data from lean mice while still on chow are depicted in Fig 3F–J. **(A)** Body weights throughout the experiment. Dashed line visualizes high fat diet start. **(B)** Relative mRNA expression of LCN13 in liver 17 wk after AAV administration. **(C, D, E)** Blood glucose, (D) plasma insulin levels, and (E) HOMA-IR from 5-h fasted animals 8 and 14 wk after AAV administration. **(F)** IPGTT (2 g/kg D-glucose) after 5 h of fasting 8 and 14 wk after AAV administration. **(G)** IPITT (1.2 and 1.5 U/kg insulin, respectively) after 5 h of fasting 9 and 15 wk after AAV administration. **(H)** Cholesterol, triglyceride, high-density lipoprotein, and low-density lipoprotein levels in serum 17 wk after AAV administration. Data information: All data are presented as mean ± SEM. *$P \leq 0.05$; ****$P \leq 0.0001$. For both IPGTT and IPITT two-way ANOVA (Tukey's test) and for all other data sets, one-way ANOVA (Tukey's test) was used.

**Figure 6.  Hepatic LCN13 overexpression does not affect key parameters of glucose and lipid metabolism in db/db mice.**

PBS or adeno-associated viruses (AAVs) overexpressing either LCN13 (LCN) or a mutated, untranslated GFP (Ctrl) were administered to *db/db* male mice (B6.BKS(D)-*Lepr^db*/J, 7 wk) at a dose of 5 × 10^11 vg/mouse via the tail vein (n = 8–9 mice for each group). C57BL/6J wild type mice (7 wk) were used as further control (wt). **(A)** Relative mRNA expression of LCN13 in liver 15 wk after AAV administration. **(B)** Representative immunoblot of LCN13 in plasma 4 wk after AAV administration. **(C)** ALT and AST levels in serum 15 wk after AAV administration. **(D)** Weight of body and liver 15 wk after AAV administration. **(E)** Blood glucose and HbA1c levels in ad-libitum fed animals 15 wk after AAV injection. **(F)** Blood glucose, plasma insulin levels, and HOMA-IR from 16-h fasted animals 10 wk after AAV administration. **(G)** IPGTT (1 g/kg D-glucose) after 16 h

et al, 2018), and some can be bound by the same receptor (Boudjelal et al, 1996; Flower, 2000). Therefore, a difference in relative expression levels of such related lipocalins between this and previous studies could explain the inability to reproduce former data. By analyzing expression levels of related lipocalins in mice with experimentally manipulated LCN13 levels in our study, we excluded the possibility that a compensatory regulation of these lipocalins obscured LCN13's metabolic importance (Fig 8).

The aforementioned functional redundancy together with the possibility of both rather specific and promiscuous ligand binding capacities (Schlehuber & Skerra, 2005) already suggests that the lipocalins family represent a rather complex and intertwined system. Indeed, LCN13 would not be the first lipocalin whose function is controversially discussed in the literature. The most intriguing example is LCN2, as the metabolic phenotype of the respective knockout mice was distinct in three independent laboratories: Law et al (2010) reported that LCN2 deficiency protects mice from obesity-induced insulin resistance, whereas Guo et al (2010) showed LCN2-dependent promotion of metabolic dysfunction (Guo et al, 2010; Law et al, 2010). In a third study, LCN2 did not affect diet-induced insulin resistance (Jun et al, 2011). Guo et al (2010) and Jun et al (2011) used the same genetic approach to generate LCN2 knockout mice (deletion of exon 2–5), whereas Law applied a different strategy (deletion of exons 1–6). However, all studies validated the LCN2 knockout by the absence of circulating LCN2. Apart from the generation of truncated or differentially spliced variants of LCN2 because of these distinct approaches, Jun et al (2011) also discuss differences in diet, husbandry, and gut microbiota as underlying factors for the varying phenotypes of these studies.

In light of these results and the fact that we could solely establish LCN13's insulinotropic and proliferation inducing potential in an isolated in vitro system, but not in vivo, we postulate that, like LCN2, LCN13 is part of a rather delicate and complex system. Subtle changes in diet, husbandry, and gut microbiota could negatively affect the availability of its cargo as well as the tissue-responsiveness toward LCN13.

In summary, our data question the importance of LCN13 as a systemic metabolic regulator, which clearly diminishes its therapeutic potential. Nevertheless, if the aforementioned confounding factor(s) was/were identified, the knowledge of LCN13's general insulinotropic potential, next to its already published insulin-sensitizing capacity, could prove useful in developing a new therapeutic strategy for insulin resistant individuals.

# Materials and Methods

### Recombinant viruses

For long-term hepatic overexpression, the full-length LCN13 cDNA (NCBI Reference Sequence: NM_153558.1) was cloned into the recombinant viral backbone plasmid pdsAAV2-LP1 established

previously (Rose et al, 2011). The control vector encodes for a mutated GFP (pdsAAV2-LP1-GFPmut) whose translation is inhibited because of inserted stop codons. AAVs were subsequently produced by Vigene Biosciences using the described recombinant viral backbone vectors, the pDGΔVP helper plasmid (Grimm et al, 1998), and the mutated p5E18-VD2/8 expression vector (Gao et al, 2002) which encodes for AAV2 rep and a mutated cap protein. Unless indicated otherwise, 5 × $10^{11}$ viral particles were injected per mouse via the tail vein.

### Nanoparticles

A LCN13-targeted siRNA (sense strand [5′-3′]: cuGuGAGAAuAAuaGcu-cAdTsdT, antisense strand [5′-3′]: UGAGCuAUuAUUCUcAcAGdTsdT; A, C, G, U = RNA nucleotides, dT = DNA nucleotide, a, c, u = 2′-O-methylated nucleotides, s = phosphorothioate linkages) was selected, synthesized, and packaged into hepatocyte-specific lipo-nanoparticles (LNP) by Axolabs GmbH. As control, LNP carrying a siRNA targeting luciferase were used. For hepatic knockdown of LCN13, mice were injected with 0.5 mg/kg LNP via the tail vein.

### Recombinant LCN13 production

To generate recombinant mammalian LCN13 protein, the LCN13 cDNA (NCBI Reference Sequence: NM_153558.1) lacking the putative signal sequence (amino acid 1–18) was cloned into the SP-6xHis-Fc-pEFIRES-PURO vector, carrying the signaling peptide of a known secreted protein (kind gift of Dr Anastasia Georgiadi) to generate SP-6xHis-Fc-LCN13. Both 6xHis-Fc-LCN13 and the control 6xHis-Fc were produced in CHO-S cells (kindly provided by Dr Anastasia Georgiadi). In short, the cells were transfected using lipofectamine 3000 reagent (Life Technologies) and successfully transfected cells were selected with 25 μg/ml puromycin (Sigma-Aldrich) in DMEM/F12 culture medium (Life Technologies) supplemented with 10% FBS in a humidified incubator containing 5% $CO_2$ and 21% $O_2$. For cell expansion, stable CHO-S cells were cultured in suspension in 100 ml serum-free OptiCHO medium (Life Technologies) supplemented with 0.5 mg/ml alanyl-glutamine (Sigma-Aldrich) at 120 rpm (New Brunswick S41i CO2 [Eppendorf]) and 37°C for 5–7 d. Afterwards, cells were further grown in 300 ml of the same medium at 120 rpm and 31°C for another 5 d. After 10–12 d, the cell suspension was centrifuged at 9,500$g$ and 4°C for 15 min. The supernatant was supplemented with 1M NaCl, 5 mM imidazole, and 1× Halt Protease Phosphatase Inhibitor-Cocktail (EDTA-free; Life Technologies), passed through a Corning bottle-top vacuum filter with a pore size of 0.45 μm (Sigma-Aldrich), degassed and adjusted to a pH of 7.2 on ice. For protein purification, HisTrap HP 5 ml chromatography columns (GE Healthcare) were attached to a ÄKTA pure 25 (GE Healthcare). Columns were washed with 50 ml ultrapure water and equilibrated with 50 ml washing buffer 1 (W1; 0.2 M NaCl, 0.02 M NaHPO$_4$, and 5 mM imidazole, pH 7.2). Supernatants were loaded at a flow rate of 1 ml/min onto the columns, the system was washed

of fasting 10 wk after AAV administration. **(H)** IPITT (2 U/kg insulin) after 5 h of fasting 14 wk after AAV administration. **(I)** Cholesterol, triglyceride, high-density lipoprotein, and low-density lipoprotein levels in serum 15 wk after AAV injection. Data information: All data are presented as mean ± SEM. ***$P$ ≤ 0.001, ****$P$ ≤ 0.0001. For both IPGTT and IPITT two-way ANOVA (Tukey's test) and for all other data sets one-way ANOVA (Tukey's test) was used. Source data are available for this figure.

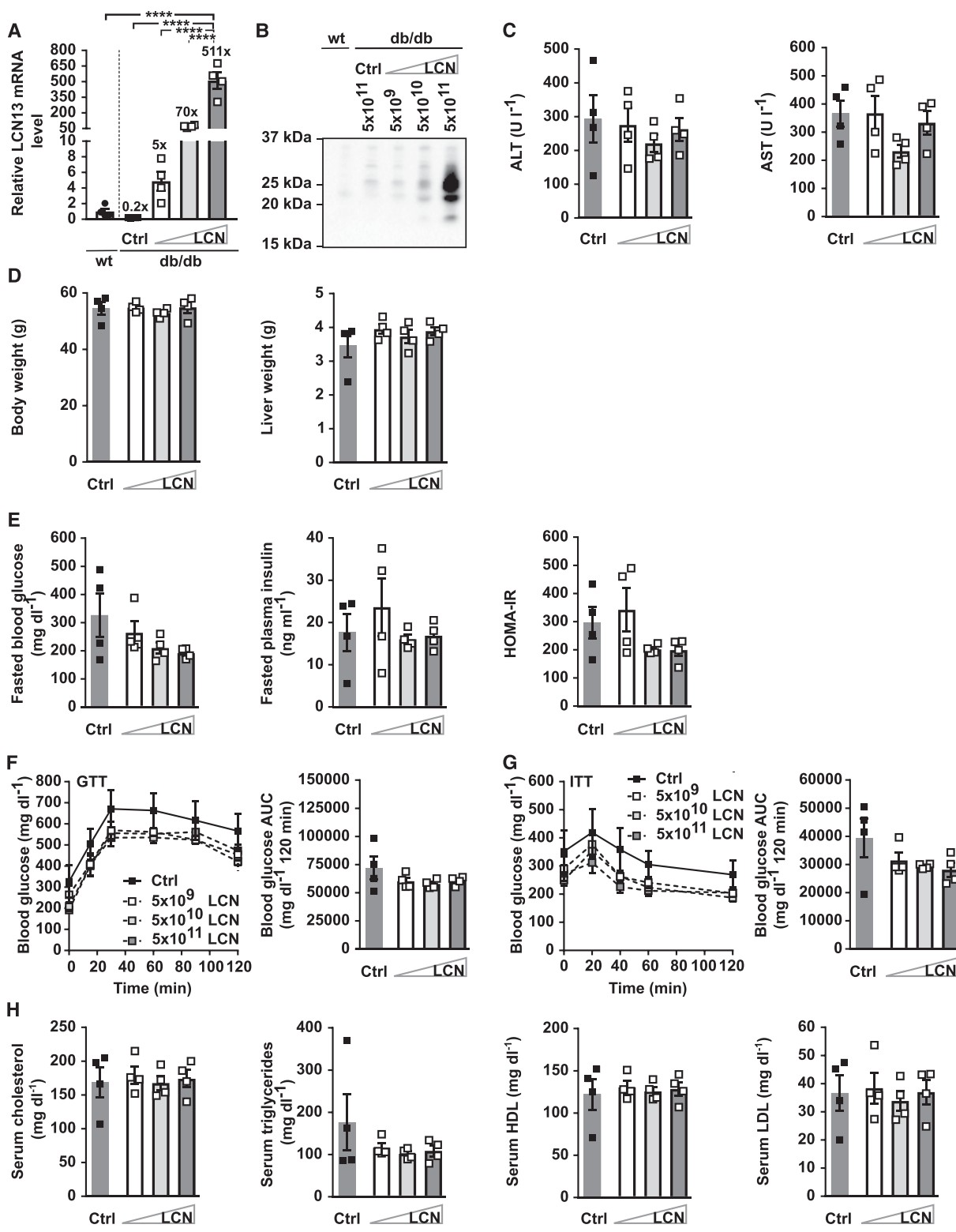

**Figure 7. Viral dosage does not affect the metabolic potential of LCN13.**
PBS or adeno-associated viruses (AAVs) overexpressing either LCN13 (LCN) or a mutated, untranslated GFP (Ctrl) were administered to *db/db* male mice (B6.BKS(D)-*Lepr$^{db}$*/J, 7 wk) at a dose of 5 × 10$^9$, 5 × 10$^{10}$, or 5 × 10$^{11}$ vg/mouse via the tail vein (n = 4 mice for each group). C57BL/6J wild-type mice (7 wk) were used as further control (wt). **(A)** Relative mRNA expression of LCN13 in liver 6 wk after AAV administration. **(B)** Representative immunoblot of LCN13 in plasma 3 wk after AAV administration. **(C)** Alanine transaminase (ALT) and aspartate transaminase (AST) in serum 6 wk after AAV administration. **(D)** Weight of body and liver 6 wk after AAV administration. **(E)** Blood glucose, plasma insulin levels, and HOMA-IR from 5-h fasted animals 4 wk after AAV administration. **(F)** IPGTT (1 g/kg D-glucose) after 5 h of fasting 4 wk after AAV

with 50 ml W1 and subsequently with 50 ml washing buffer 2 (W2; 0.2 M NaCl, 0.02 M NaHPO₄, and 20 mM imidazole, pH 7.2) at a flow rate of 2 ml/min. Protein was eluted in seven fractions of 5 ml each using 400 mM imidazole elution buffer (0.2 M NaCl and 0.02 M NaHPO₄, pH 7.2) at a flow rate of 1 ml/min. Eluted protein was dialyzed in DPBS (Life Technologies) and snap frozen until further usage.

### LC–MS analysis of Fc-LCN13

Dialyzed Fc-LCN13 and the Fc control (10 and 12 μg protein, respectively) were proteolyzed by a modified filter-aided sample preparation protocol as described previously (Grosche et al, 2016). Briefly, proteins were diluted 1:10 with 0.1 M Tris/HCl, pH 8.5, and 50 μl 100 mM dithiothreitol was added for 30 min at 60°C. After cooling down, 500 μl UA buffer (8 M urea and 1 M Tris–HCl, pH 8.5, diluted in HPLC-grade water) and 100 μl 300 mM iodoacetamide were added and incubated for 30 min at RT in the dark. Eluates were transferred to 30 kD cutoff centrifuge filters (Sartorius) and washed five times with 200 μl UA-buffer and two times with 100 μl ABC buffer (50 mM NH₃HCO₃ diluted in HPLC-grade water). After washing, proteins were subjected to proteolysis at RT for 2 h with 0.5 μg Lys C in 40 μl ABC-buffer followed by addition of 1 μg trypsin and incubation at 37°C overnight. Peptides were collected by centrifugation and acidified with 0.5% trifluoroacetic acid. Acidified eluted peptides were analyzed in the data-dependent mode on a Q Exactive HF mass spectrometer (Thermo Fisher Scientific) online coupled to a Ultimate 3000 RSLC nano-HPLC (Dionex) as described previously (Lepper et al, 2018). Samples were automatically injected and loaded onto the C18 trap column, eluted after 5 min, and separated on the C18 analytical column (Acquity UPLC M-Class HSS T3 column, 1.8, 75 μm × 250 mm; Waters) by a 90 min non-linear acetonitrile gradient at a flow rate of 250 nl/min. MS spectra were recorded at a resolution of 60,000 and after each MS1 cycle, the 10 most abundant peptide ions were selected for fragmentation. Acquired raw data were loaded into Progenesis QI software for proteomics for MS1 intensity based label-free quantification (v3.0, Nonlinear Dynamics; Waters), and analyzed as described previously (Grosche et al, 2016). MSMS spectra were exported and searched against the SwissProt mouse database (16,868 sequences) using the Mascot search engine (version 2.5.1). Search settings were: enzyme trypsin, 10 ppm peptide mass tolerance and 0.02 D fragment mass tolerance, one missed cleavage allowed, carbamidomethylation was set as fixed modification, methionine oxidation, and asparagine and glutamine de-amidation were allowed as variable modifications. A Mascot-integrated decoy database search was performed with an average false discovery rate of <1%. Peptide assignments were re-imported into the Progenesis QI software. The abundances of all unique peptides allocated to each protein were summed up. The resulting normalized abundances of the individual proteins were used for calculation of fold-changes of protein ratios between Fc-LCN13 and the Fc control samples. Only proteins quantified by at least two unique peptides and identified by three or more spectral counts were considered.

### Animal experiments

Animal handling was conducted within national and European Union ethical guidelines and experiments were approved by the state ethics committee and government of Upper Bavaria (licenses ROB-55.2-2532.Vet_02-15-164 and ROB-55.2-2532.Vet_02-17-49). Unless indicated otherwise, mice were maintained under specific pathogen-fee conditions on a 12 h light–dark cycle and fed ad libitum with regular rodent chow diet (Ref # 1314; Altromin Spezialfutter GmbH & Co. KG) and free access to water. Male C57Bl6/N mice were obtained from Charles River Laboratories which were used in the therapeutic HFD study and for the LCN13 tissue expression screen as well as from Janvier Laboratories which were used in the preventive HFD and the knock-down study. Male db/db mice (B6.BKS(D)-Leprdb/J [000697] homozygote) and respective wild-type controls (C57BL/6J [000664]) were purchased from the Jackson Laboratory. For HFD studies, a HFD (Ref # D12492i, 20% protein, 20% carbohydrate, 60% fat in kcal) and a control low-fat diet (Ref # D12450Ji, 20% protein, 70% carbohydrate, 10% fat in kcal) were obtained from Research Diets. In the therapeutic HFD study, 5-wk-old C57Bl6/N mice were kept on HFD for 14 wk before AAVs were administered. In the preventive HFD study, 7-wk-old C57Bl6/N mice were injected with AAV before they were fed the HFD 5 wk later. db/db mice received AAV at the age of 7 wk. For LCN13 knockdown, C57Bl6/N mice were injected with LNP at the age of 8 wk. To study β-cell proliferation in vivo, mice were i.p. injected with 100 mg/kg BrdU (Sigma-Aldrich) on three consecutive days, before the studies were terminated. At the end of an experiment, mice were killed by cervical dislocation. Trunk blood was collected in micro tubes 1.1 ml Z-Gel (SARSTEDT AG & Co. KG) for preparation of serum. Organs were collected, weighed, and snap-frozen in liquid nitrogen. Serum and tissues were stored at –80°C until further analyses.

### Assessment of glucose and insulin tolerance

For glucose tolerance tests, mice were fasted overnight (16 h: from 6 PM to 10 AM) or for 5 h (from 8 AM to 1 PM) and injected intraperitoneally with 1 or 2 g/kg D-glucose. In detail, db/db mice were fasted for 16 h before injected with 1 g/kg D-glucose. db/db mice which received different doses of AAV were fasted for 5 h and injected with 1 g/kg D-glucose. C57Bl6/N mice of the therapeutic HFD study received 1 g/kg D-glucose after they were fasted for 16 h. In the preventive HFD study, C57Bl6/N mice were fasted for 5 h and injected with 2 g/kg D-glucose. In the LCN13 knockdown study, C57Bl6/N mice were also fasted for 5 h and received 2 g/kg D-glucose. Blood glucose was monitored before and 15, 30, 60 and 120 min after glucose injection with an Accu-Chek Performa glucometer (Roche) or in case of blood glucose levels above 600 mg/dl with the StatStrip Xpress Glucose Meter from Nova Biomedical. Blood samples were collected from the tail vein using EDTA-covered capillary

administration. **(G)** IPITT (3 U/kg insulin) after 5 h of fasting 5 wk after AAV administration. **(H)** Cholesterol, triglyceride, high-density lipoprotein and low-density lipoprotein in serum 6 wk after AAV administration. Data information: All data are presented as mean ± SEM. ****P ≤ 0.0001. For both IPGTT and IPITT two-way ANOVA (Tukey's test) and for all other data sets one-way ANOVA (Tukey's test) was used.
Source data are available for this figure.

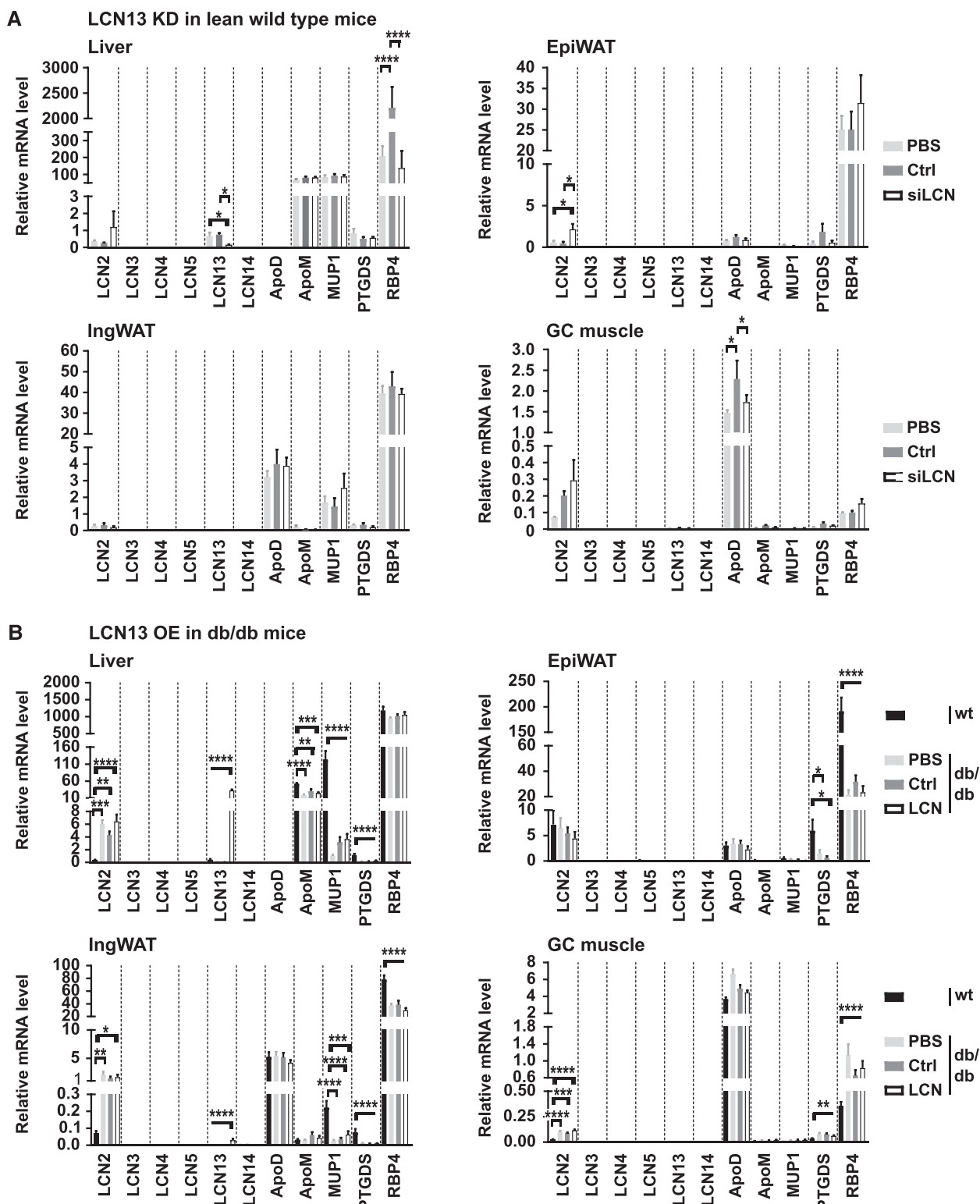

**Figure 8. Manipulation of LCN13 levels does not trigger compensatory regulation of related lipocalin family members.**
**(A)** 0.5 mg/kg LNPs containing siRNA either against LCN13 (siLCN) or luciferase (siC) were i.v. injected in C57BL/6N male mice (8 wk; n = 8 mice for each group). Relative mRNA expression of Lcn2, Lcn3, Lcn4, Lcn5, Lcn13, Lcn14, apolipoprotein D (ApoD), apolipoprotein M (ApoM), major urinary protein 1 (Mup1), prostaglandin-H2 D-isomerase (Ptgds), and retinol binding protein 4 (Rbp4) in the liver, epididymal white adipose tissue (EpiWAT), inguinal white adipose tissue (IngWAT), and gastrocnemius (GC) muscle 2 wk after LNP administration. **(B)** PBS or adeno-associated viruses overexpressing either LCN13 (LCN) or an untranslated GFP (Ctrl) were administered to db/db male mice (B6.BKS(D)-Lepr^db/J, 7 wk) at a dose of 5 × 10^11 vg/mouse via the tail vein (n = 8–9 mice for each group). C57BL/6J wild type mice (7 wk) were used as further control (wt). Relative mRNA expression of lipocalin family members 15 wk after adeno-associated virus administration. Data information: All data are presented as mean ± SEM. *P ≤ 0.05, **P ≤ 0.01, ***P ≤ 0.001, ****P ≤ 0.0001. For all data, an outlier testing (ROUT; Q = 1%) was performed. Cleaned data were analyzed by one-way ANOVA (Tukey's test).

tubes (SARSTEDT AG & Co. KG). Plasma insulin was measured using the mouse insulin enzyme-linked immunosorbent assay (ELISA) kit from ALPCO. The homeostasis model assessment (HOMA) index was calculated as fasting blood glucose (mmol/l) × fasting plasma insulin ($\mu$IU/ml)/22.5. For insulin tolerance tests, mice were fasted for 5 h (from 8 AM to 1 PM) and intraperitoneally injected with 1, 1.2, 1.5, 2, or 3 U/kg human insulin (Huminsulin Normal; Lilly). In detail, db/db mice were injected with 2 U/kg insulin. db/db mice which received different doses of AAV were injected with 3 U/kg insulin. C57Bl6/N mice of the therapeutic HFD study received 1 and 1.5 U/kg insulin, whereas C57Bl6/N mice of the preventive HFD study received 1.2 and 1.5 U/kg insulin. In the LCN13 knockdown study, C57Bl6/N mice received 1 U/kg insulin. Tail vein blood glucose was measured before and 20, 40, 60, 90, and 120 min after insulin injection.

## Serum analysis

Cholesterol, triglyceride, high-density lipoprotein, low-density lipoprotein, alanine transaminase (ALT), aspartate aminotransferase (AST), and HbA1c levels were measured on the AU480 chemistry analyzer (Beckman Coulter) using commercially available kits (Beckman Coulter).

## Islet morphometric analysis

Dissected pancreatic samples were fixed in 4% (wt/vol) neutrally buffered formalin and embedded in paraffin. Islet analysis was performed on two sections of 3 $\mu$m which were ~300 $\mu$m apart. After a co-staining for insulin (monoclonal rabbit anti-insulin, #3014; Cell Signaling; Alexa Fluor 750–conjugated anti-rabbit, A21039; Invitrogen), for glucagon (polyclonal guinea pig anti-glucagon, M182; Takara; Alexa-Fluor647-conjugated anti guinea pig, A21450; Invitrogen), and for BrdU (biotinylated anti-BrdU, ab2284; Abcam; CY3 conjugated streptavidin, SA1010; Invitrogen) nuclei were labeled with Hoechst 33342 (Thermo Fisher Scientific). The stained tissue sections were scanned with an AxioScan.Z1 digital slide scanner (Zeiss) equipped with a 20× magnification objective. Quantification of insulin, glucagon or BrdU positive cells were performed by using image analysis software Definiens Developer XD2 (Definiens) following a previously published procedure (Feuchtinger et al, 2015). The calculated parameters were the amount of $\beta$-cells (insulin-positive cell area) and $\alpha$-cells (glucagon-positive cell area) as percentage of the total islet area. In addition, the amount of proliferating $\beta$-cells (BrdU and insulin double-positive cell area as percentage of the total insulin-positive cell area) or proliferating $\alpha$-cells (BrdU and glucagon double-positive cell area as percentage of the total glucagon-positive cell area) was determined.

## MIN6 pseudo-islets

For pseudo-islet formation, $2 \times 10^5$ Min6-m9 cells (kind gift of Dr Susumu Seino of Kobe University; Minami, Yano et al, 2000) were seeded in 2 ml per well in Corning Costar Ultra-Low Attachment 6 Well Plates (Sigma-Aldrich). Cells were cultured in DMEM (Life Technologies) supplemented with 20% FBS, 1% penicillin, 1% streptomycin, 0.01% $\beta$-mercaptoethanol (Sigma-Aldrich), and 1× glutamine (Life Technologies) under shaking (70 rpm) at 37°C in a

humidified incubator containing 5% $CO_2$ and 21% $O_2$. After 3 d, pseudo-islets were used for experiments.

## Isolation of primary mouse islets for GSIS

Primary islets were isolated from male C57Bl6/N mice (Charles River Laboratories) at the age of 8–12 wk. After cervical dislocation, pancreata were inflated via the pancreatic duct with 3 ml collagenase P solution (1 mg collagenase P [Roche]/mL Hank's Balanced Salt Solution [HBSS; Sigma-Aldrich]) supplemented with 1% protease-free BSA (SERVA). Pancreas digestion was conducted in a 37°C warm water bath for 15 min. To stop collagenase action, the digest was washed with ice-cold HBSS medium supplemented with 1% BSA and all further steps were conducted on ice. Large tissue debris were removed by passing the digest through a 500-$\mu$m cell strainer (pluriSelect). In the following, islet cells were separated using a Histopaque density gradient (45.5% Histopaque-1077 and 54.5% Histopaque-1119 [v:v]; Sigma-Aldrich). Islets at the interface were removed, washed twice with HBSS medium supplemented with 1% BSA and subsequently transferred into islet culture medium (RPMI 1640 [Life Technologies] supplemented with 10% FBS [Thermo Fisher Scientific], 1% penicillin and 1% streptomycin [Life Technologies]). After 1–2 h of incubation at 37°C in a humidified incubator containing 5% $CO_2$ and 21% $O_2$, islets were hand-picked into new islet culture medium. Islets were allowed to recover overnight, before they were used for experiments.

## GSIS of MIN6 pseudo-islets and primary islets

Primary islets as well as pseudo-islets were washed in Krebs–Ringer buffer (KRB; 2.54 mM $CaCl_2$, 10 mM Hepes, 4.74 mM KCl, 1.19 mM $KH_2PO_4$, 118.5 mM NaCl, 25 mM $NaHCO_3$, and 1.19 mM $MgSO_4$, supplemented with 0.1% BSA [RIA grade; Sigma-Aldrich] with low glucose [2.8 mM]). Afterwards, 10 islets or pseudo-islets of comparable size were incubated in 300 $\mu$l KRB with low glucose (2.8 mM) for 1 h at 37°C in a humidified incubator containing 5% $CO_2$ and 21% $O_2$. Subsequently, islets and pseudo-islets were transferred into 300 $\mu$l KRB buffer at 2.8 or 16.7 mM glucose with different treatments for 1 or 2 h, respectively. 10 nM Exendin-4 (Ex-4; Sigma-Aldrich) served as positive control, whereas PBS treatment was used as negative control. To study LCN13's insulinotropic potential islet and pseudo-islets were treated with 10 nM of the commercially available recombinant bacterial LCN13 (R&D Systems). To test the bioactivity of our in-house produced Fc-LCN13, pseudo-islets were treated with 20 nM of the respective protein or the Fc control. Secretion medium was collected, centrifuged (1,000 rpm at 4°C for 1 min) and stored at –20°C until further analysis. For insulin content, islet or pseudo-islets were transferred and sonicated in 150 $\mu$l ice-cold acid-ethanol solution (70% ethanol supplemented with 0.18 M HCl; three pulses of 30 Hz for 1 s each). Insulin was measured using the ELISA kit from ALPCO according to the manufacturer's instructions.

## Generation of primary islet cell monolayer and ex vivo proliferation assay

For details about husbandry, islet isolation, and the preparation of islet cell monolayers see Kluth et al (2015). Islet cells were isolated

from male C57Bl6/J mice (Charles River Laboratories) on regular chow diet (SNIFF) at the age of 18–20 wk. After seeding, islet cells were maintained as monolayers in RPMI 1640 supplemented with 10% FBS, 1% penicillin, 1% streptomycin, and 11 mM glucose for 24 h. Subsequently, cells were further grown in fresh medium supplemented with 1:1,000 (v:v) FibrOut (CHI Scientific) for 3 d. To assess LCN13's potential to induce islet cell proliferation, monolayers were cultured in the same medium without FibrOut in the presence of 100 µM BrdU (Sigma-Aldrich) and either 200 nM recombinant bacterial LCN13 (R&D Systems) or PBS for 4 d with a medium change after 2 d. Afterwards, cells were fixed with 4% paraformaldehyde and immunofluorescence staining and analysis of BrdU incorporation was performed as described in Kluth et al (2015).

### Western blot

For detection of circulating LCN13, plasma collected in EDTA-covered capillary tubes (SARSTEDT AG & Co. KG) was diluted 1:20 and 10 µl (~24 µg protein) were loaded on Novex 12% or 4–20% tris-glycine mini gels (Life Technologies) and separated by SDS–PAGE, followed by transfer onto nitrocellulose membranes (Bio-Rad Laboratories). For total protein detection, gels were stained with Coomassie Brilliant Blue R 250 (Sigma-Aldrich). Membranes were immunoblotted with an antibody specific for mouse LCN13 (AF7974; R&D Systems; [Ustunel et al, 2016]) and the corresponding secondary antibody (sc-2924; Santa Cruz Biotechnology). For detection of the recombinant mammalian Fc-LCN13 and the Fc control, the HRP-conjugated rabbit anti-human IgG H&L (ab6759; Abcam) was used. Proteins were detected using SuperSignal West Femto Chemilumineszenz-Substrat (Thermo Fisher Scientific), the Chem-iDoc XRS+ imaging system (Bio-Rad Laboratories) and Image Lab (Bio-Rad Laboratories).

### Quantitative Taqman RT-PCR

Total RNA was extracted from homogenized mouse tissues using TRIzol Reagent (Life Technologies). cDNA was prepared using the Quantitect Reverse Transcription Kit (QIAGEN). mRNA abundance was measured using the TaqMan probe-based qPCR (Life Technologies) and the Quantstudio 6 quantitative PCR machine (Life Technologies). Following primer probes were used: Apolipoprotein D (ApoD, Mm01342307_m1), apolipoprotein M (ApoM, Mm00444525_m1), LCN2 (Mm01324470_m1), LCN3 (Mm00440138_m1), LCN4 (Mm03048210_m1), LCN5 (Mm00468329_m1), major urinary protein 1 (Mup1, Mm03647538_g1), odorant binding protein 2a (Obp2a/LCN13, Mm00463685_m1), odorant binding protein 2b (Obp2b/LCN14, Mm01328294_g1), peroxisome proliferator-activated receptor γ (PPAR-γ, Mm00440940_m1), prostaglandin D2 synthase (PTGDS, Mm01330613_m1), retinol binding protein 4 (RBP4, Mm00803264_g1), and stearoyl-CoA desaturase-1 (Scd1, Mm00772290_m1). mRNA abundances were normalized to TATA-box binding protein mRNA levels (Mm01277042_m1).

### Tissue lipid extraction

Hepatic lipid extraction was conducted as described before (Folch et al, 1957). Total cholesterol (Randox Laboratories Ltd.), non-esterified fatty acid (FUJIFILM Wako Chemicals), phospholipid (FUJIFILM Wako Chemicals), and triglyceride (Sigma-Aldrich) levels were determined using commercially available kits according to the manufacturer's instructions.

### Statistical analysis

Data are shown as mean ± SEM. Statistical analysis was performed using Graph Pad Prism v. 7. For two groups, non-repeated measures, and equal variances (F-test: $P ≥ 0.05$) an unpaired, two-tailed $t$ test was used. In one-factorial designs with more than two groups, one-way ANOVA and for multifactorial study designs, two-way ANOVA was applied. Dunnett's, Sidak's, or Turkey's multiple comparisons tests were performed when significant differences were calculated. $P ≤ 0.05$ was considered statistically significant.

## Data Availability

The full-length LCN13 cDNA was retrieved from the National Institute of Health (NIH) genetic sequence database GenBank (Accession: NM_153558.1 | https://www.ncbi.nlm.nih.gov/nuccore/NM_153558). The MS data from this publication have been deposited to the PRIDE database (https://www.ebi.ac.uk/pride/) and assigned the identifier PXD022682.

## Supplementary Information

## Acknowledgements

We thank Luke Harrison for editing the manuscript and Susumu Seino for kindly providing Min6-m9 cells. This work was supported by the Collaborative Research Center 1118 "Reactive metabolites and diabetic complications" awarded to P Nawroth and S Herzig.

### Author Contributions

L Bühler: data curation, investigation, and methodology.
A Maida: investigation.
ES Vogl: investigation.
A Georgiadi: investigation.
A Takacs: investigation.
O Kluth: investigation.
A Schuermann: conceptualization.
A Feuchtinger: investigation.
C von Toerne: investigation.
F-F Tsokanos: investigation and methodology.
K Klepac: investigation and methodology.
G Wolff: investigation and methodology.
M Sakurai: investigation and methodology.
B Ekim-Uestuenel: investigation and methodology.
P Nawroth: conceptualization.

S Herzig: conceptualization, data curation, funding acquisition, and writing—original draft, review, and editing.

## Conflict of Interest Statement

The authors declare that they have no conflict of interest.

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
