## [Reviewer comments · Life Science Alliance]

Life Science Alliance

Lipocalin 13 enhances insulin secretion but is dispensable for systemic metabolic control

Stephan Herzig, Lea Bühler, Adriano Maida, Elena Vogl, Anastasia Georgiadi, Andrea Takacs, Oliver Kluth, Annette Schuermann, Annette Feuchtinger, Christine von Toerne, Foivos Tsokanos, Katarina Klepac, Gretchen Wolff, Minako Sakurai, Bilgen Ekim-Uestueneel, and Peter Nawroth

DOI: <https://doi.org/10.26508/lsa.202000898>

Corresponding author(s): Stephan Herzig, Helmholtz Center Munich

Review Timeline:

Submission Date:	2020-08-29
Editorial Decision:	2020-10-02
Revision Received:	2021-01-02
Editorial Decision:	2021-01-06
Revision Received:	2021-01-12
Accepted:	2021-01-12

Scientific Editor: Shachi Bhatt

Transaction Report:

October 2, 2020

Re: Life Science Alliance manuscript #LSA-2020-00898-T

Prof. Stephan Herzig
Helmholtz Center Munich
Institute for Diabetes and Cancer
Ingolstaedter Landstraße 1
Neuherberg 85764
Germany

Dear Dr. Herzig,

Thank you for submitting your manuscript entitled "Lipocalin 13 enhances insulin secretion but is dispensable for systemic metabolic control" to Life Science Alliance (LSA). The manuscript has been reviewed by the editors and outside referees (reviewer comments below). As you will see, the reviewers were enthusiastic about the study and its potential impact, but have raised some concerns that should be addressed to further strengthen manuscript for publication at LSA. Therefore, although we are unable to publish the current version of the manuscript, we encourage you to submit a revised manuscript that addresses all of the reviewers' concerns

We would be happy to discuss the individual revision points further with you, should this be helpful. A revised manuscript may be re-reviewed, most likely by some or all of the original referees. When submitting the revision, please include a letter addressing the reviewers' comments point-by-point and a copy of the text with alterations highlighted (boldfaced or underlined). The typical time frame for revisions is three months. In an effort to expedite the review process, papers are generally considered through only one revision cycle.

While you are revising your manuscript, please also attend to the below editorial points to help expedite the publication of your manuscript.

Please use the link below to log in to your account and submit your revised manuscript
<https://lsa.msubmit.net/cgi-bin/main.plex>

Thank you for considering Life Science Alliance (LSA) as an appropriate venue for your research. Please reach out to me if you have any questions.

Sincerely,

Shachi Bhatt, Ph.D.
Executive Editor
Life Science Alliance
<https://www.life-science-alliance.org/>
Tweet @SciBhatt @LSAjournal

- A letter addressing the reviewers' comments point by point.
- An editable version of the final text (.DOC or .DOCX) is needed for copyediting (no PDFs).
- High-resolution figure, supplementary figure and video files uploaded as individual files: See our detailed guidelines for preparing your production-ready images, <https://www.life-science-alliance.org/authors>
- Summary blurb (enter in submission system): A short text summarizing in a single sentence the study (max. 200 characters including spaces). This text is used in conjunction with the titles of papers, hence should be informative and complementary to the title and running title. It should describe the context and significance of the findings for a general readership; it should be written in the present tense and refer to the work in the third person. Author names should not be mentioned.

B. MANUSCRIPT ORGANIZATION AND FORMATTING:

Reviewer #1 (Comments to the Authors (Required)):

The paper presents a very thorough investigation into the potential effects of lipocalin 13 on glucose homeostasis in mice. Using a variety of methods, no effect of lipocalin 13 overexpression or silencing on various measures of glucose homeostasis is found. The data presented in the paper, although overwhelmingly negative, are novel and are helpful to further narrow down the functional role of lipocalin 13. The data conflict with three recent papers, one of which by the same authors, in which a role of lipocalin 13 in glucose homeostasis is demonstrated. The paper presents a balanced appraisal of the new data in relation to the published literature.

Specific comments are listed below.

The constant switching from regular figures to supplementary figures disrupts the reading and flow of the paper. Authors are encouraged to move more data to the regular figures. In addition, the latter part of the results section (almost 2 pages!) only refers to supplementary figures. If these data are so important that they deserve nearly two pages of description, they shouldn't be relegated to the supplementary data. This is another very good reason to move much more data to the regular figures.

The first figure presents data indicating that lipocalin stimulates insulin secretion. However, these findings are not discussed in the discussion section. The authors should discuss the potential physiological relevance of these findings, taking into consideration that insulin levels, glucose levels, glucose tolerance, and insulin tolerance were not affected by Lipocalin 13 in vivo. Specifically, how can the stimulatory effect of lipocalin 13 on insulin secretion be reconciled with lack of effect of hepatic overexpression or silencing on glucose homeostasis. If the data are not physiologically meaningful, why should they be included in the paper?

How did the authors arrive at the concentrations of recombinant Lipocalin 13 that were used in the studies with pancreatic islets. Do these concentrations match the physiological plasma concentration?

The title of figure 2 should be corrected for grammar.

The suggestion is to move supplementary figure 1B to figure 2.

Do the authors have an idea why Fc alone also stimulated insulin secretion (Sup figure 2D)

It seems that the dose of insulin used in the insulin tolerance tests was insufficient, as blood glucose hardly went down upon insulin injection (fig 2F, 4C). Any possible stimulatory effect of lipocalin 13 in insulin resistance will be missed in these experiments. This should probably be addressed in the discussion. By contrast, glucose went down much more in figure 5C.

How is it possible that glucose tolerance is similar between Ctrl and PBS condition (figure 4b, right panel 14 weeks). Also, insulin tolerance is similar (Figure 4c, right panel). It thus seems that the high fat feeding did not cause insulin resistance in these animals.

Reviewer #2 (Comments to the Authors (Required)):

The group of Herzig aims at the vigorous pre-clinical evaluation of LCN13's therapeutic potential for metabolic disorders, specifically diabetes. Previous data suggested that LCN13 may be a prime candidate for clinical development for the treatment of diet associated metabolic disorders. By applying state of the art loss- and gain-of-function approaches in vivo, Bühler et al show that while LCN13 triggers glucose-dependent insulin secretion and cell proliferation in primary mouse islets, the effect cannot be translated into the in vivo setting.

Therefore, these data clearly contrast with previously published data and call into question a therapeutic role of LCN13 in diabetes and its related disorders.

The strength of the study is the vigorous evaluation of the effect of LCN13 in metabolic disease using state of the art loss- and gain-of-function approaches. The manuscript is elegantly written and the conclusions are supported by the data. I have only minor comments:

- In the introduction the authors state that the liver is the central metabolic organ. It may be more appropriate to say that the liver is an important metabolic organ.
- In the first paragraph of the results section, the authors state that LCN13 robustly induced GSIS in

and ex vivo. Yet, at this stage, in vivo experiments are not shown.

- In lean B16 mice with targeted hepatic KD of LCN13, LCN13 levels in other organs should be displayed, but more importantly, circulating levels of LCN13 in KD vs control.

- In section 'Hepatocyte-borne LCN13 does not affect systemic energy homeostasis in mice with diet-induced obesity' it is stated that a curative approach was chosen. Reading the manuscript the approach was rather of therapeutic nature.

- In the discussion, the authors state that systemic energy homeostasis was assessed. While it is true that body weight was measured other important parameters of energy homeostasis such as energy expenditure, food intake and fecal and urinary energy loss are not reported. Thus, instead of systemic energy homeostasis, systemic lipid and glucose metabolism seems to be a more appropriate term.

Reviewer #3 (Comments to the Authors (Required)):

In this paper the authors have examined the ability of lipocalin 13 to modulate systemic glucose homeostasis using in vivo approaches. They report that although lipocalin 13 is able to improve insulin secretion in vitro it fails to have significant impact in regulating systemic glucose homeostasis in vivo. These data are at odds with previous report and the authors discuss the various possibilities for the different outcomes. They conclude by questioning the relevance of lipocalin 13 as a potential therapeutic for obesity and diabetes.

The experimental models are well planned and the results are convincing and conclusion is justified.

Minor comment:

The immunostaining in Fig 1 D would benefit from co-immunostaining of BrDu with insulin and glucagon separately to show the effects on proliferation.

Point-by-point response

We highly appreciate the referee's constructive and supportive comments.

Please find our point-by-point rebuttal below.

Reviewer #1 (Comments to the Authors (Required)):

The paper presents a very thorough investigation into the potential effects of lipocalin 13 on glucose homeostasis in mice. Using a variety of methods, no effect of lipocalin 13 overexpression or silencing on various measures of glucose homeostasis is found. The data presented in the paper, although overwhelmingly negative, are novel and are helpful to further narrow down the functional role of lipocalin 13. The data conflict with three recent papers, one of which by the same authors, in which a role of lipocalin 13 in glucose homeostasis is demonstrated. The paper presents a balanced appraisal of the new data in relation to the published literature.

Specific comments are listed below.

The constant switching from regular figures to supplementary figures disrupts the reading and flow of the paper. Authors are encouraged to move more data to the regular figures. In addition, the latter part of the results section (almost 2 pages!) only refers to supplementary figures. If these data are so important that they deserve nearly two pages of description, they shouldn't be relegated to the supplementary data. This is another very good reason to move much more data to the regular figures.

We fully agree with Reviewer #1. For that reason, we moved data validating LCN13 knockdown/overexpression to the main figures. Whenever it was possible, additional data from the supplementary figures were moved to the main figures without compromising the graph sizes. Supplementary figures 8 and 9 were completely converted to main figures (Figure 7 and Figure 8). By that, the revised version is comprised of 5 supplementary figures instead of 9.

The first figure presents data indicating that lipocalin stimulates insulin secretion. However, these findings are not discussed in the discussion section. The authors should discuss the potential physiological relevance of these findings, taking into consideration that insulin levels, glucose levels, glucose tolerance, and insulin tolerance were not affected by Lipocalin 13 in vivo. Specifically, how can the stimulatory effect of lipocalin 13 on insulin secretion be reconciled with lack of effect of hepatic overexpression or silencing on glucose homeostasis. If the data are not physiologically meaningful, why should they be included in the paper?

Especially in light of previous papers discussing controversial roles of other lipocalin family members in systemic metabolism, our hypothesis is that lipocalin 13 is also part of such a delicate system in which little changes could affect its importance/metabolic regulatory potential. We think that such little changes could be stemming from differences in (a) background inflammation, (b) diet, especially lipid components which could serve as potential cargo of LCN13, (c) mouse husbandry and (d) gut microbiota. All these potential confounding factors could compromise LCN13's actions in vivo, whereas it still shows its effects in an isolated in vitro system. We agree with Reviewer #1, that we should comment more specifically on the discrepancy between our in vitro and in vivo data in the discussion section. Therefore, we included an additional paragraph at the end of the discussion. If the aforementioned confounding factor(s) was/were identified, the knowledge of LCN13's general insulinotropic potential, next to its already published insulin-sensitizing capacity, could prove useful in developing a new therapeutic strategy for insulin resistant individuals. Therefore we think these data are an important part of the manuscript.

How did the authors arrive at the concentrations of recombinant Lipocalin 13 that were used in the studies with pancreatic islets. Do these concentrations match the physiological plasma concentration?

The concentration used in our in vitro studies were chosen based on a pharmacological point of view, rather than on the actual physiological plasma concentration. The used concentrations were chosen based on previous publications which demonstrated insulin sensitizing potential of 1 - 200 nM recombinant LCN13 in 3T3-L1 adipocytes and C2C12 myotubes (Cho et al., 2011; Ekim Ustunel et al., 2016). In our hands, 10 nM bacterial recombinant LCN13 was sufficient to robustly enhance glucose-dependent insulin secretion in and ex vivo. For our in-house produced mammalian recombinant LCN13, we chose a concentration of 20 nM, as, in contrast to the bacterial protein, it showed two distinct sizes in western blot analysis, most probably due to posttranslational modifications. By increasing the concentration to 20 nM, we ensured that both "versions" of LCN13 were present at a concentration of 10 nM. For the ex vivo proliferation assay 200 nM proved to be most effective.

Quantification of LCN13 in mouse plasma/serum is difficult, because functional ELISA kits are not available. We tested the "ELISA Kit for Odorant Binding Protein 2A (OBP2A)", product number: SEC686Mu from Cloud-Clone which gave only non-conclusive results. In humans, lean non-diabetic males and females have an average physiological molar LCN13 plasma concentration of (0.1 ± 0.06) nM and (0.08 ± 0.06) nM, respectively ($n = 8$ male and 4 female subjects).

The title of figure 2 should be corrected for grammar.

The title of figure 2 was corrected.

The suggestion is to move supplementary figure 1B to figure 2.

We thank Reviewer #1 for this valid suggestion. Based on this comment, we decided to move all data validating LCN13 knockdown/overexpression to the main figures.

Do the authors have an idea why Fc alone also stimulated insulin secretion (Sup figure 2D).

Both (6xHis-)Fc-LCN13 and the (6xHis-)Fc control were produced in CHO-S cells and the His-tagged recombinant proteins present in the secretion media were purified using high-performance immobilized metal affinity chromatography (IMAC) columns. We could validate the identity and enrichment of LCN13 by both coomassie-stained protein gel/Western Blot and mass spectrometry (Figure S2B and C, Table S1). The mass spectrometry data showed that 88 % of the total normalized abundances are derived from LCN13 indicating that there are other proteins still present after the applied purification procedure. Therefore, we cannot exclude the presence of other insulinotropic substances in both the Fc-LCN13 and the Fc control samples. As both protein samples were generated in the same system in parallel, we think that the Fc control with its “background insulinotropic potential” is the best negative control to use in this experiment.

It seems that the dose of insulin used in the insulin tolerance tests was insufficient, as blood glucose hardly went down upon insulin injection (fig 2F, 4C). Any possible stimulatory effect of lipocalin 13 in insulin resistance will be missed in these experiments. This should probably be addressed in the discussion. By contrast, glucose went down much more in figure 5C.

We thank Reviewer #1 for this valid comment. We addressed this issue in the text at the corresponding sections (Fig 3J, 4G and 6H).

How is it possible that glucose tolerance is similar between Ctrl and PBS condition (figure 4b, right panel 14 weeks). Also, insulin tolerance is similar (Figure 4c, right panel). It thus seems that the high fat feeding did not cause insulin resistance in these animals.

We think this was a misunderstanding by the referee. We would like to clarify that control mice on LFD are NOT shown in the graphs, except for Fig 4A and B. All mice receiving intravenous injections, regardless of the substance (PBS, Ctrl and LCN13 overexpressing AAVs), were fed a 60% HFD. We decided to solely show the HFD control mice in the graphs as they are the “proper” controls for any potential LCN13 effect in obese mice and by this the interpretation of the graphs is more straight forward.

Reviewer #2 (Comments to the Authors (Required)):

The group of Herzig aims at the vigorous pre-clinical evaluation of LCN13's therapeutic potential for metabolic disorders, specifically diabetes. Previous data suggested that LCN13 may be a prime candidate for clinical development for the treatment of diet associated metabolic disorders. By applying state of the art loss- and gain-of-function approaches in vivo, Bühler et al show that while LCN13 triggers glucose-dependent insulin secretion and cell proliferation in primary mouse islets, the effect cannot be translated into the in vivo setting.

Therefore, these data clearly contrast with previously published data and call into question a therapeutic role of LCN13 in diabetes and its related disorders.

The strength of the study is the vigorous evaluation of the effect of LCN13 in metabolic disease using state of the art loss- and gain-of-function approaches. The manuscript is elegantly written and the conclusions are supported by the data. I have only minor comments:

We appreciate the referee's positive comments on our manuscript.

In the introduction the authors state that the liver is the central metabolic organ. It may be more appropriate to say that the liver is an important metabolic organ.

We thank Reviewer #2 for this valid comment. We exchanged “the central metabolic organ” with “an important metabolic organ”.

In the first paragraph of the results section, the authors state that LCN13 robustly induced GSIS in and ex vivo. Yet, at this stage, in vivo experiments are not shown.

We agree with Reviewer #2 that it would be nice to complement the in vitro part about LCN13's insulinotropic effect with in vivo data. Therefore, we added data showing insulin levels in fasted lean mice 15 minutes after i.p. glucose injection. In order to not interrupt the flow of the manuscript, we kept the order and added the mentioned data as Fig 2D and 3E and H.

In lean Bl6 mice with targeted hepatic KD of LCN13, LCN13 levels in other organs should be displayed, but more importantly, circulating levels of LCN13 in KD vs control.

We thank Reviewer #2 for this valid comment. In our opinion, showing hepatic LCN13 expression data is enough, as we identified the liver as only source of LCN13 (Fig 2A). Fig 8A validates this assumption, as we could exclude LCN13

expression in epididymal/inguinal white adipose as well in muscle tissue of LCN13 KD mice.

Quantification of LCN13 in mouse plasma/serum is difficult, because functional ELISA kits are not available. We tested the “ELISA Kit for Odorant Binding Protein 2A (OBP2A)”, product number: SEC686Mu from Cloud-Clone which gave only non-conclusive results. We additionally tried to validate the LCN13 knockdown in the circulation by western blot. Even though the used LCN13 antibody LCN13 (AF7974, R&D Systems) was raised in sheep, the secondary antibody (sc-2924, Santa Cruz Biotechnology) cross-reacted with the endogenous IgG light chain. As the IgG light chain migrates within the molecular weight range of LCN13 (25 kDa), it was impossible to clearly identify the less abundant endogenous LCN13 levels. Therefore, we tried to circumvent this cross-reaction by using the VeriBlot for IP Detection Reagent (HRP; ab131366, abcam) which does not bind to reduced endogenous antibodies, as it only recognizes native, non-reduced ones. Unfortunately, the VeriBlot reagent didn't recognize our primary antibody (two different concentrations of the VeriBlot reagent were used: 1:500 and 1:2000). As another strategy, we depleted mouse serum from albumin and IgG antibodies using the Proteome Purify 2 Mouse Serum Protein Immunodepletion Resin (MIDR002, R&D Systems). The depletion technically worked well (after two rounds of depletion, no albumin was detectable by western blot). However, the only clearly detected band did not seem to represent LCN13 as its intensity did not decrease upon LCN13 knockdown, albeit the knockdown was validated at the mRNA level. In conclusion it seems that the used primary anti-LCN13 antibody works well with samples containing excess amount of LCN13 (recombinant protein injections/LCN13 overexpression AAV; see Figure 3A and F, 4B, 6B and 7B), but not with the lower endogenous LCN13 levels.

In section 'Hepatocyte-borne LCN13 does not affect systemic energy homeostasis in mice with diet-induced obesity' it is stated that a curative approach was chosen. Reading the manuscript the approach was rather of therapeutic nature.

We thank Reviewer #2 for this comment and amended the manuscript by exchanging the term “curative” with “therapeutic”.

In the discussion, the authors state that systemic energy homeostasis was assessed. While it is true that body weight was measured other important parameters of energy homeostasis such as energy expenditure, food intake and fecal and urinary energy loss are not reported. Thus, instead of systemic energy homeostasis, systemic lipid and glucose metabolism seems to be a more appropriate term.

We thank Reviewer #2 for this valid comment and amended the manuscript accordingly.

Reviewer #3 (Comments to the Authors (Required)):

In this paper the authors have examined the ability of lipocalin 13 to modulate systemic glucose homeostasis using in vivo approaches. They report that although lipocalin 13 is able to improve insulin secretion in vitro it fails to have significant impact in regulating systemic glucose homeostasis in vivo. These data are at odds with previous report and the authors discuss the various possibilities for the different outcomes. They conclude by questioning the relevance of lipocalin 13 as a potential therapeutic for obesity and diabetes.

The experimental models are well planned and the results are convincing and conclusion is justified.

We thank the referee for his/her support.

Minor comment:

The immunostaining in Fig 1 D would benefit from co-immunostaining of BrDu with insulin and glucagon separately to show the effects on proliferation.

When the data was generated, we did not do an insulin co-staining as the majority of islet cells are insulin-producing β -cells (in our in vivo data ~73%, see Fig S5B). However, we fully agree with Reviewer #3 that the data would gain value from BrdU and insulin/glucagon co-staining. Unfortunately, as the shown slides stem from primary cell monolayers, we cannot re-stain the slides for insulin and glucagon.

References

Cho, K.W., Zhou, Y., Sheng, L., and Rui, L. (2011). Lipocalin-13 regulates glucose metabolism by both insulin-dependent and insulin-independent mechanisms. *Mol Cell Biol* 31, 450-457.

Ekim Ustunel, B., Friedrich, K., Maida, A., Wang, X., Krones-Herzig, A., Seibert, O., Sommerfeld, A., Jones, A., Sijmonsma, T.P., Sticht, C., *et al.* (2016). Control of diabetic hyperglycaemia and insulin resistance through TSC22D4. *Nature communications* 7, 13267.

January 6, 2021

RE: Life Science Alliance Manuscript #LSA-2020-00898-TR

Prof. Stephan Herzig
Helmholtz Center Munich
Institute for Diabetes and Cancer
Ingolstaedter Landstraße 1
Neuherberg 85764
Germany

Dear Dr. Herzig,

Thank you for submitting your revised manuscript entitled "Lipocalin 13 enhances insulin secretion but is dispensable for systemic metabolic control". We would be happy to publish your paper in Life Science Alliance pending final revisions necessary to meet our formatting guidelines.

Along with the points listed below, please also attend to the following,

- please make sure the author order in your manuscript and our system match and that there is a name discrepancy between the manuscript file and the system
- please use the [10 author names, et al.] format in your references (i.e. limit the author names to the first 10)
- please add callouts for Figures S3D and S5A
- please add a scale bar for the last panel (labelled 'outlier') in Figure S5A

A. FINAL FILES:

- An editable version of the final text (.DOC or .DOCX) is needed for copyediting (no PDFs).
- High-resolution figure, supplementary figure and video files uploaded as individual files: See our detailed guidelines for preparing your production-ready images, <https://www.life-science-alliance.org/authors>
- Summary blurb (enter in submission system): A short text summarizing in a single sentence the

study (max. 200 characters including spaces). This text is used in conjunction with the titles of papers, hence should be informative and complementary to the title. It should describe the context and significance of the findings for a general readership; it should be written in the present tense and refer to the work in the third person. Author names should not be mentioned.

B. MANUSCRIPT ORGANIZATION AND FORMATTING:

Sincerely,

Shachi Bhatt, Ph.D.
Executive Editor
Life Science Alliance
<https://www.lsjournal.org/>
Tweet @SciBhatt @LSAJournal

January 12, 2021

RE: Life Science Alliance Manuscript #LSA-2020-00898-TRR

Prof. Stephan Herzig
Helmholtz Center Munich
Institute for Diabetes and Cancer
Ingolstaedter Landstraße 1
Neuherberg 85764
Germany

Dear Dr. Herzig,

Thank you for submitting your Research Article entitled "Lipocalin 13 enhances insulin secretion but is dispensable for systemic metabolic control". It is a pleasure to let you know that your manuscript is now accepted for publication in Life Science Alliance. Congratulations on this interesting work.

DISTRIBUTION OF MATERIALS:

Again, congratulations on a very nice paper. I hope you found the review process to be constructive and are pleased with how the manuscript was handled editorially. We look forward to future exciting submissions from your lab.

Sincerely,

Shachi Bhatt, Ph.D.

Executive Editor

Life Science Alliance

<https://www.lsjournal.org/>
